# IL-12 reprograms CAR-expressing natural killer T cells to long-lived Th1-polarized cells with potent antitumor activity

Elisa Landoni[1], Mark G. Woodcock [1,2], Gabriel Barragan[3], Gabriele Casirati [4,5], Vincenzo Cinella [4,5], Simone Stucchi[1], Leah M. Flick[1], Tracy A. Withers[1], Hanna Hudson[1], Giulia Casorati [6], Paolo Dellabona [6], Pietro Genovese [4,5], Barbara Savoldo[1,7], Leonid S. Metelitsa[3] & Gianpietro Dotti [1,8] ✉

Human natural killer T cells (NKTs) are innate-like T lymphocytes increasingly used for cancer immunotherapy. Here we show that human NKTs expressing the pro-inflammatory cytokine interleukin-12 (IL-12) undergo extensive and sustained molecular and functional reprogramming. Specifically, IL-12 instructs and maintains a Th1-polarization program in NKTs in vivo without causing their functional exhaustion. Furthermore, using CD62L as a marker of memory cells in human NKTs, we observe that IL-12 maintains long-term CD62L-expressing memory NKTs in vivo. Notably, IL-12 initiates a de novo programming of memory NKTs in CD62L-negative NKTs indicating that human NKTs circulating in the peripheral blood possess an intrinsic differentiation hierarchy, and that IL-12 plays a role in promoting their differentiation to long-lived Th1-polarized memory cells. Human NKTs engineered to co-express a Chimeric Antigen Receptor (CAR) coupled with the expression of IL-12 show enhanced antitumor activity in leukemia and neuroblastoma tumor models, persist long-term in vivo and conserve the molecular signature driven by the IL-12 expression. Thus IL-12 reveals an intrinsic plasticity of peripheral human NKTs that may play a crucial role in the development of cell therapeutics.

Adoptive transfer of CAR-T cells achieved remarkable results in several hematological malignancies, but lacks robust objective clinical responses in patients with solid tumors. The modest activity of CAR-T cells in solid tumors can be attributed to the lack of ideal antigen targets, the presence of physical barriers for immune cell trafficking and penetration as well as tumor immune suppressive mechanisms[1]. Multiple engineering strategies have been proposed to improve the therapeutic effects of CAR-T cells in solid tumors. Alternatively, other immune cell subsets have been proposed as a cellular platform for CAR engineering such as NK cells[2,3], γδ-T cells[4], and Vα24-invariant natural killer T cells (NKTs)[5] that possess specific immune properties distinct from conventional αβTCR T cells.

Type-1 NKTs are an evolutionarily conserved sub-lineage of innate-like T-cells[6,7]. In humans, NKTs are characterized by the

[1]Lineberger Comprehensive Cancer Center, University of North Carolina, Chapel Hill, NC, USA. [2]Division of Oncology, Department of Medicine, University of North Carolina, Chapel Hill, NC, USA. [3]Center for Advanced Innate Cell Therapy, Texas Children's Cancer Center, Department of Pediatrics, Baylor College of Medicine, Houston, TX, USA. [4]Dana-Farber/Boston Children's Cancer and Blood Disorder Center, Boston, USA. [5]Harvard Medical School, Boston, USA. [6]Experimental Immunology Unit, Division of Immunology, Transplantation and Infectious Diseases, IRCCS San Raffaele Scientific Institute, Milan, Italy. [7]Department of Pediatrics, University of North Carolina, Chapel Hill, NC, USA. [8]Department of Microbiology and Immunology, University of North Carolina, Chapel Hill, NC, USA. ✉e-mail: gdotti@med.unc.edu

expression of an invariant TCR α-chain TRAV10 + TRAJ18+ (iTCR, formerly Vα24-Jα18), which is preferentially paired with the TRBV25-1 (formerly Vβ11) TCR β-chain[8], and by reactivity to alpha-galactosylceramide (αGalCer) presented by the monomorphic non-classical MHC molecule CD1d[9,10]. Human NKTs are known to infiltrate solid tumors conferring improved outcomes in several human malignancies[11–13]. We thus developed CAR-redirected human NKTs (CAR-NKTs) to target solid tumors. Specifically, we demonstrated that the infusion of CAR-NKTs can promote tumor regression in neuroblastoma models[14] and in children with relapsed or resistant neuroblastoma[15,16].

Our clinical study indicates that human CAR-NKTs benefit from the co-expression of the IL-15 cytokine to support their expansion and persistence in cancer patients[15,16]. Furthermore, we have identified that human CAR-NKTs expressing CD62L show superior proliferative and persistence capacity and antitumor activity compared to CAR-NKTs lacking CD62L expression[15,17] suggesting that these innate immune cells may undergo a differentiation process resembling the central-memory differentiation of αβTCR T cells. Importantly, CD62L expression in CAR-NKTs correlates with better clinical outcome in patients. Leveraging the intrinsic plasticity of human peripheral NKTs may thus lead to the generation of long-lived CAR-NKTs with high potential of controlling tumor growth long-term.

IL-12 is a heterodimeric cytokine produced by dendritic cells, macrophages, and B cells in response to microbial pathogens and exert potent pro-inflammatory and tumor-suppressive activity[18]. NKs and NKTs express the IL-12Rβ1 and IL-12Rβ2 receptors and become pro-inflammatory cells in response to IL-12[19].

Here we show that the transgenic expression of IL-12 in human NKTs and CAR-NKTs not only imprints a Th1 polarization, but causes an extensive molecular reprogramming of NKTs that leads to the generation of exhaustion-resistant long-lived memory cells with potent antitumor activity.

## Results

### NKTs expressing IL-12 are Th1-polarized

Human NKTs were isolated from the peripheral blood, expanded, and transduced as previously described[20] (Fig. 1a). After isolation and activation, NKTs were transduced at day 5 of culture with a gamma retroviral vector encoding either a control vector encoding GFP (GFP) or the p40 and p35 subunits of IL-12 connected by a flexible linker and GFP (IL12(I)GFP) (Fig. 1b). The transduction efficiency (Fig. 1c and Fig. S1A) and purity (Fig. 1d and Fig. S1B) of NKTs evaluated by measuring the expression of GFP and iTCR were superior to 70% and 90%, respectively. IL-12 was constitutively produced by IL12(I)GFP NKTs, and was detectable in the supernatant (Fig. 1e). IL-12 engages the IL-12 receptor (CD212) in NKTs and causes its downregulation in IL12(I)GFP NKTs (Fig. 1f and Fig. S1C). The autocrine/paracrine IL-12/CD212 interaction is functional in NKTs because we observed the phosphorylation of STAT4, which is involved in the IL-12R signaling pathway[21] (Fig. 1g). IL-12 signaling has been associated with the production of IFN-γ in NK and NKTs and Th1-polarization[19,22]. We thus analyzed the relative production of IFN-γ and IL-4 by human NKTs upon stimulation[5]. IL12(I)GFP NKTs showed superior release of IFN-γ and reduced release of IL-4 compared to control GFP NKTs (Fig. 1h). The increased production of IFN-γ by IL12(I)GFP NKTs was associated with an increase in the IFN-γ mRNA transcripts (Fig. 1i). Accordingly, we found upregulation of the transcription factor T-bet in IL12(I)GFP NKTs, which is associated with IFN-γ transcription[23,24] (Fig. 1i). We also assessed if IL-12 expression promotes the generation of polyfunctional NKTs. NKTs were transduced with retroviral vectors in which GFP was switched with ΔNGFR to allow sorting of the transduced NKTs[25], and processed to assess polyfunctionality at single-cell resolution. Activated IL12-expressing NKTs showed significantly higher polyfunctionality defined as the percentage of cells that secrete two or more cytokines (Fig. 1j

and Fig. S1D). Moreover, IL-12 expressing NKTs showed elevated polyfunctional strength index (PSI, a combination of the poly-functionality with the strength of cytokine secretion), compared to control NKTs, and displayed significantly higher effector and inflammatory PSI (Fig. S1E). Overall, these data indicate that NKTs expressing IL-12 are polyfunctional Th1-polarized cells, and that IL-12 has an autocrine/paracrine mode of action.

### IL-12 reprograms NKTs to activation and proliferation rather than exhaustion

To fully characterize the transcriptomic effects of IL-12 in NKTs, we compared IL12(I)GFP and GFP NKTs by performing RNASeq. We observed the increased expression of *IL-12A/B* genes introduced by the retroviral vector in the IL12(I)GFP NKTs, and the differential expression of approximately 380 genes between IL12(I)GFP and GFP NKTs (Fig. 2a and Fig. S2A), including upregulation of markers of activation/ exhaustion such as *PD-1*, *HAVRC2 (TIM3)*, *LAG-3* and *CTLA4*, but also markers associated with memory NKTs such as *SELL* (CD62L), and the transcription factor *LEF1*[26]. We then compared published gene signature expression scores for cellular pathways and T-cell exhaustion between IL12(I)GFP and GFP NKTs. Two significant gene sets related to T-cell exhaustion (GSE9650 and GSE41867) correlated with the signature of IL12(I)GFP NKTs, but neither demonstrated a large log-fold change between the two sample groups. However, we found significant overexpression of PD-1, TIM-3, LAG-3, and CTLA4 in IL12(I)GFP NKTs by flow cytometry compared to GFP NKTs (Fig. 2b and Fig. S2B), while the typical composition in CD4+, CD8+, and CD4-CD8- NKT subsets remained unaffected (Fig. S2C). IL12(I)GFP NKTs and GFP NKTs showed similar functional capacity in vitro in eliminating CD1d+ target cells loaded with αGalCer (Fig. 2c and Fig. S2D, E). In sharp contrast, IL12(I)GFP NKTs showed increased proliferative capacity upon stimulation compared to GFP NKTs as assessed by the CTV dilution assay by flow cytometry (Fig. 2d). In line with their superior proliferative capacity and overexpression of *SELL* mRNA, IL12(I)GFP NKTs showed high expression of CD62L by flow cytometry compared to GFP NKTs (Fig. 2e), which is consistent with our previous observation that CD62L expression in human NKTs identifies a cell subset with high proliferative capacity[17]. We confirmed that CD62L expression in IL12(I)GFP NKTs is regulated at the transcriptional level because qPCR demonstrated mRNA upregulation of both CD62L and FOXO1, which is a transcription factor implicated in the transcriptional regulation of CD62L in murine and human cells[27] (Fig. 2f). To further demonstrate that IL-12 initiates a transcriptional program in NKTs, which includes de novo expression of CD62L, we sorted CD62L+ and CD62L- NKTs from peripheral blood and exposed them to either recombinant IL-2 alone or IL-12 and IL-2 in vitro (Fig. 2g). IL-12 increased the CD62L expression in sorted CD62L+ NKTs and a fraction of CD62L- NKTs acquired de novo CD62L expression, while IL-2 did not affect CD62L expression (Fig. 2g). NKTs expressing CD62L de novo upon exposure to IL-12 showed similar proliferative capacity as NKTs expressing CD62L without IL-12 exposure (Fig. S2F). Finally, we compared the effects of IL-12, IL-15 and IL-21[28] when expressed via genetic engineering in NKTs (Fig. S3A). Engineered NKTs showed the same rate of apoptosis, and IL-12 did not cause increased apoptosis in NKTs compared to IL-15 and IL-21 (Fig. S3B). Notably, only IL-12 promoted superior release of IFN-γ and reduced release of IL-4 upon stimulation (Fig. S3C). Overall, these data indicate that IL-12 initiates a unique de novo transcriptional program in NKTs that induces Th1 polarization, but also expression of markers associated with memory formation.

### NKTs expressing IL-12 acquire long-term longevity in vivo

Having observed that IL12(I)GFP NKTs are Th1-polarized and have enhanced activation and proliferation in vitro, we evaluated the engraftment of IL12(I)GFP NKTs in vivo in non-tumor-bearing NSG mice (Fig. 3a). IL12(I)GFP NKTs showed superior expansion and

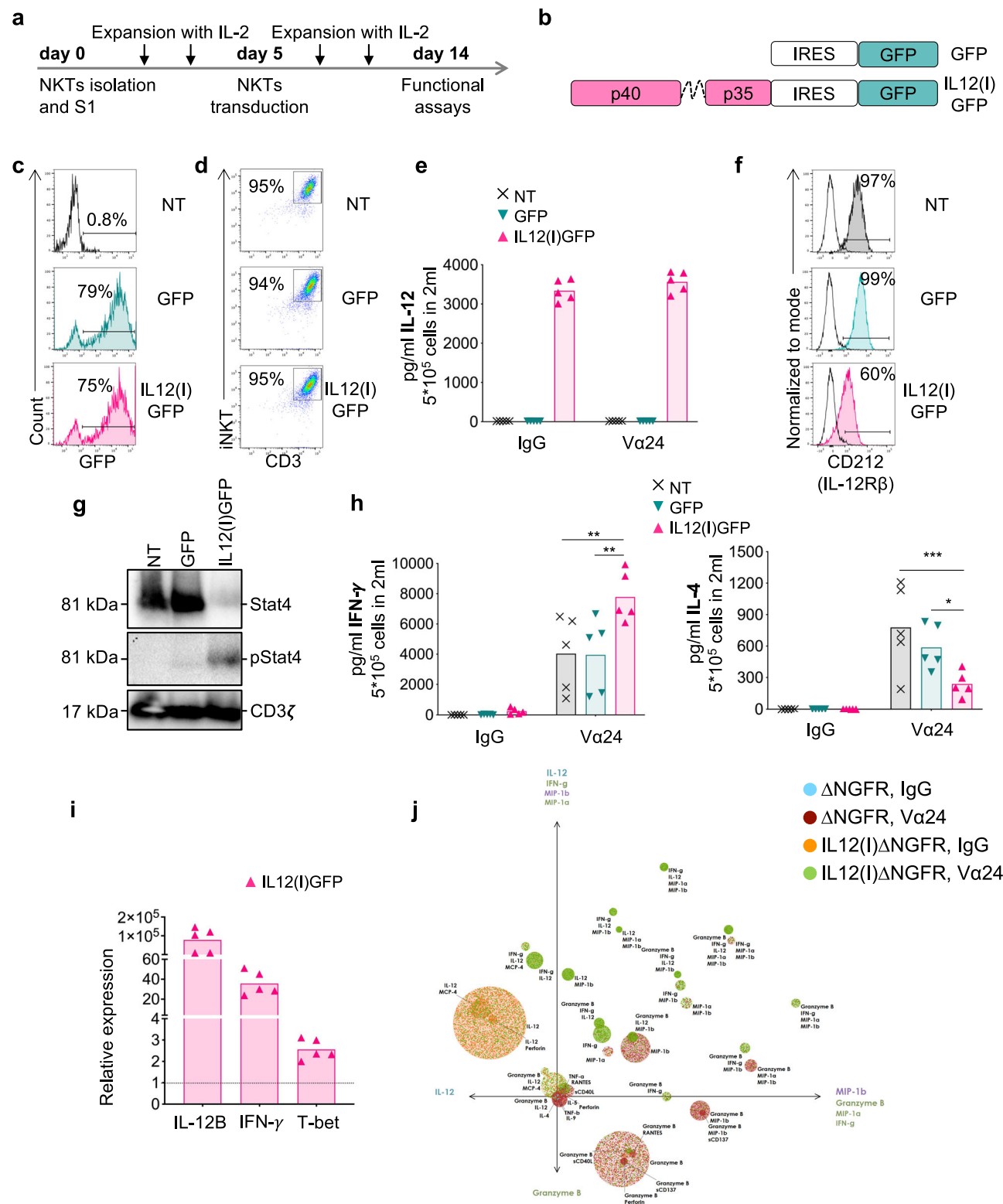

persistence in vivo in the peripheral blood, spleen, and liver of NSG mice up to day 30 (Fig. 3b), and maintained high CD62L expression (Fig. 3c). At day 30, IL12(I)GFP NKTs were selected from the spleen and injected in new non-tumor-bearing NSG mice to mimic a serial transplant. IL12(I)GFP NKTs showed consistent engraftment in the serial transplant (Fig. S4A, B) and maintained CD62L expression (Fig. S4C). Mice infused with IL12(I)GFP NKTs did not show signs of graft versus host disease and remained healthy. Finally, we compared the molecular signature of long-term in vivo persisting IL12(I)GFP NKTs with

that of the ex vivo expanded IL12(I)GFP or control NKTs before the infusion in mice. We observed that the expression of a set of genes including *IFNG*, *HAVCR2*, *IL12A*, *IL12B*, and *IL10* was attenuated compared to ex vivo expanded IL12(I)GFP, but remained higher than control NKTs. Interestingly, a smaller subset of genes including *SELL*, *ETV7*, and *SNCA* maintained high expression comparable to the ex vivo expanded IL12(I)GFP (Fig. 3d). Considering the role of CD62L in identifying NKTs with high proliferative capacity, we performed experiments on the gain and loss of function for CD62L in NKTs. Specifically,

**Fig. 1 | NKTs expressing IL-12 acquire pro-inflammatory properties. a** Schematic timeline of the protocol used to select, transduce, and expand human NKTs obtained from peripheral blood. **b** Schematics of the retroviral vectors used to engineer human NKTs. Gamma retroviral vectors encoding either GFP only (GFP) or the p40 and p35 subunits of IL-12 connected by a flexible linker and GFP (IL12(I)GFP). **c** Representative flow cytometry plots of NKTs showing the transduction efficiency measured as the percentage of GFP⁺ cells in non-transduced NKTs (NT) and NKTs transduced with GFP or IL12(I)GFP vectors. **d** Representative flow cytometry plots showing NKT purity measured as percentages of iNKT⁺CD3⁺ cells in NT, GFP, and IL12(I)GFP NKTs. **e** Quantification of IL-12 produced by NT, GFP, and IL12(I)GFP NKTs activated with control IgG Ab or with the iNKT Ab (Vα24). IL-12 was measured in supernatants collected 24 h after plating 5 ×10⁵ cells/well in 24 well plate in 2 mL of complete media without cytokines. Mean is shown; n = 5 donors. Source data are provided as a Source Data file. **f** Representative flow cytometry plots showing the expression of CD212 (IL-12Rβ) on the cell membrane of NT, GFP, and IL12(I)GFP NKTs at day 14 of culture. Empty lines represent the isotype control.

**g** Representative western blot illustrating the STAT4 phosphorylation in NT, GFP, and IL12(I)GFP NKTs at day 14 of culture; n = 4. Source data are provided as a Source Data file. **h** Quantification of IFN-γ and IL-4 produced by NT, GFP, and IL12(I)GFP NKTs after activation with the control IgG Ab or with the iNKT antibody (Vα24). Cytokines were measured in supernatants collected 24 h after plating 5 × 10⁵ cells/ well in 24 well plate in 2 mL of complete media without cytokines. Mean is shown; n = 5 donors; *p = 0.0286; **p = 0.0031 NT vs IL12(i)GFP; **p = 0.0025 GFP vs IL12(i) GFP; ***p = 0.0006; two-way ANOVA. Source data are provided as a Source Data file. **i** Relative expression of IL-12B, IFN-γ, and T-bet assessed by qPCR in IL12(I)GFP vs NT NKTs. Mean is shown; n = 5 donors. Source data are provided as a Source Data file. **j** NKTs were transduced with retroviral vectors in which GFP was switched with ΔNGFR to allow sorting of the transduced NKTs[25]. The panel illustrates the PAT-PCA visualization of control and IL-12-expressing NKTs after activation with the control IgG Ab or the iNKT Ab (Vα24) by IsoPlexis IsoCode Secretome. Each dot represents a single cell. NKTs were categorized into circles representing polyfunctional groups with the co-secretion of different cytokines; n = 4.

we either overexpressed CD62L in NKTs (CD62L.ΔNGFR) or knocked out CD62L by CRISPR-Cas9 in IL12(I)GFP NKTs (IL12(I)GFP KO.SELL) (Fig. 3e and Fig. S4D, E). While CD62L.ΔNGFR NKTs did not show long-term engraftment, IL12(I)GFP KO.SELL expanded and persisted in vivo (Fig. 3f and Fig. S4F) and remained CD62L negative (Fig. 3g and Fig. S4G), suggesting that CD62L in the IL12(I)GFP NKTs is a marker of a complex molecular signature induced by IL-12, but it is not directly responsible for the long-term engraftment of human NKTs in NSG mice. Of note, continuous exposure to IL-12 is required to ensure the long-term engraftment in vivo of NKTs because NKTs cultured ex vivo with recombinant IL-12 did not show long-term engraftment in vivo (Fig. S5A), while the administration of recombinant IL-12 in vivo sustained NKT cell persistence (Fig. S5B). Overall, these data indicate that the transgenic IL-12 initiates a transcriptional program in NKTs that leads to the generation of NKTs characterized by a high potential for in vivo expansion and persistence.

**Transgenic IL-12 enhances the antitumor activity of CAR-NKTs in leukemia**

Having observed the long-term in vivo persistence of IL12(I)GFP NKTs in non-tumor bearing mice, we used tumor models to assess if the simultaneous expression of IL-12 and a CAR in NKTs provides improved antitumor effects. GD2.CAR(I)IL12 and CD19.CAR(I)IL12 were successfully expressed in human NKTs (Fig. 4a, b and Fig. S6A), and promoted higher expression of CD62L compared to control NKTs and CD19.CAR-NKTs (Fig. 4c and Fig. S6B). To measure the anti-tumor activity in vitro, CD19.CAR(I)IL12 NKTs were cocultured with CD19⁺ tumor cells (BV-173 and Daudi). CD19.CAR(I)IL12 NKTs showed sustained antitumor activity in stress coculture experiments in which NKTs were repetitively exposed to tumor cells (Fig. 4d and Fig. S6C). To measure the antitumor activity of CD19.CAR(I)IL12 NKTs in vivo, NSG mice were engrafted with the leukemia cell line BV-173 (Fig. 4e). In this tumor model, mice treated with CD19.CAR(I)IL12 NKTs showed improved tumor control (Fig. 4f and Fig. S6D) that led to increased overall survival (Fig. 4g) compared to mice treated with CD19.CAR-NKTs. Of note, IL-12 NKTs persisted longer in vivo, and we detected higher numbers of NKTs in mice treated with CD19.CAR(I)IL12 NKTs in the peripheral blood at week 4 (Fig. 4h) and in the organs at sacrifice (Fig. S6E, F). Overall these data indicate that IL-12 increases the antitumor effects of CAR-NKTs targeting a liquid tumor.

**Transgenic IL-12 enhances the antitumor activity of CAR-NKTs in neuroblastoma**

To explore the effects of IL-12 in NKTs targeting solid tumors, we used the previously validated xenotransplant neuroblastoma model that predicted the clinical outcome in CAR-NKTs in patients with neuroblastoma[14,15]. For this new set of experiments, CD19.CAR(I)IL12), GD2.CAR and GD2.CAR(I)IL12 were successfully

expressed in human NKTs (Fig. 5a, b and Fig. S7A), confirming higher CD62L expression by NKTs in the presence of IL-12 (Fig. 5c and Fig. S7B). CAR-expressing NKTs were then activated through either the iTCR using the Vα24 Ab or the CAR using the 1A7 Ab that specifically activates the GD2.CAR[29], while the IgG was used as negative control. Both CD19.CAR(I)IL12 and GD2.CAR(I)IL12 NKTs constitutively released IL-12, but GD2.CAR(I)IL12 NKTs released more IFN-γ and less IL-4 compared to GD2.CAR-NKTs (Fig. S7C), confirming the Th1 polarization provided by IL-12 in NKTs. To measure the anti-tumor activity in vitro, GD2.CAR(I)IL12 NKTs were cocultured with the neuroblastoma cell line CHLA-255. GD2.CAR(I)IL12 NKTs showed superior antitumor activity in stress coculture experiments in which NKTs were repetitively exposed to neuroblastoma tumor cells (Fig. 5d, e and Fig. 7d) and increased proliferation (Fig. 5f). Moreover, GD2.CAR(I)IL12 NKTs chronically exposed to tumor cells maintained high expression of CD62L (Fig. S7E). We also quantified the level of cytokines produced during chronic exposure to tumor cells. GD2.CAR(I)IL12 NKTs showed sustained production of IL-12 and IFN-γ together with decreased production of IL-4 (Fig. 5g). We validated these results by repeating the same coculture scheme with a second neuroblastoma cell line LAN-1 (Fig. S7F−H). To measure the anti-tumor activity of GD2.CAR(I)IL12 NKTs in vivo, we used NSG mice engrafted with the neuroblastoma cell line CHLA-255 (Fig. 6a). In this tumor model, GD2.CAR(I)IL12 NKTs caused significant tumor control at day 63 after treatment compared to GD2.CAR-NKTs and continued to protect the mice from tumor growth after tumor re-challenge (Fig. 6b, c). The superior tumor control translated into improved tumor-free survival (Fig. 6d). NKTs co-expressing the CAR and IL-12, but not control GD2.CAR-NKTs, were detected in the peripheral blood at different time points after infusion (Fig. 6e and Fig. S8A, B), and in the liver and spleen at the time of euthanasia (Fig. S8C, D), and continued to express CD62L at high level (Fig. S8E). Serum cytokines were quantified four weeks after the infusion of NKTs. Human pro-inflammatory cytokines such as IL-12, IFN-γ, GM-CSF, and IL-2 were increased in mice treated with GD2.CAR(I)IL12 NKTs (Fig. 6f and Fig. S8F). Finally, we analyzed the gene expression profile of GD2.CAR(I) IL12 NKTs collected from mice clearing the tumor. We found that the molecular signature of long-term persisting GD2.CAR(I)IL12 NKTs was similar to the signature of long-term persisting IL12(I) GFP NKTs, but different from the signature of GFP NKTs expanded ex vivo indicating that the molecular signature driven by IL-12 in NKTs is not greatly altered by CAR expression and its engagement (Fig. 6g and Fig. S9A, B). Overall these data indicate that IL-12 increases the antitumor effects of CAR-NKTs in solid tumors and that IL-12 drives the molecular signature of CAR-NKTs in vivo.

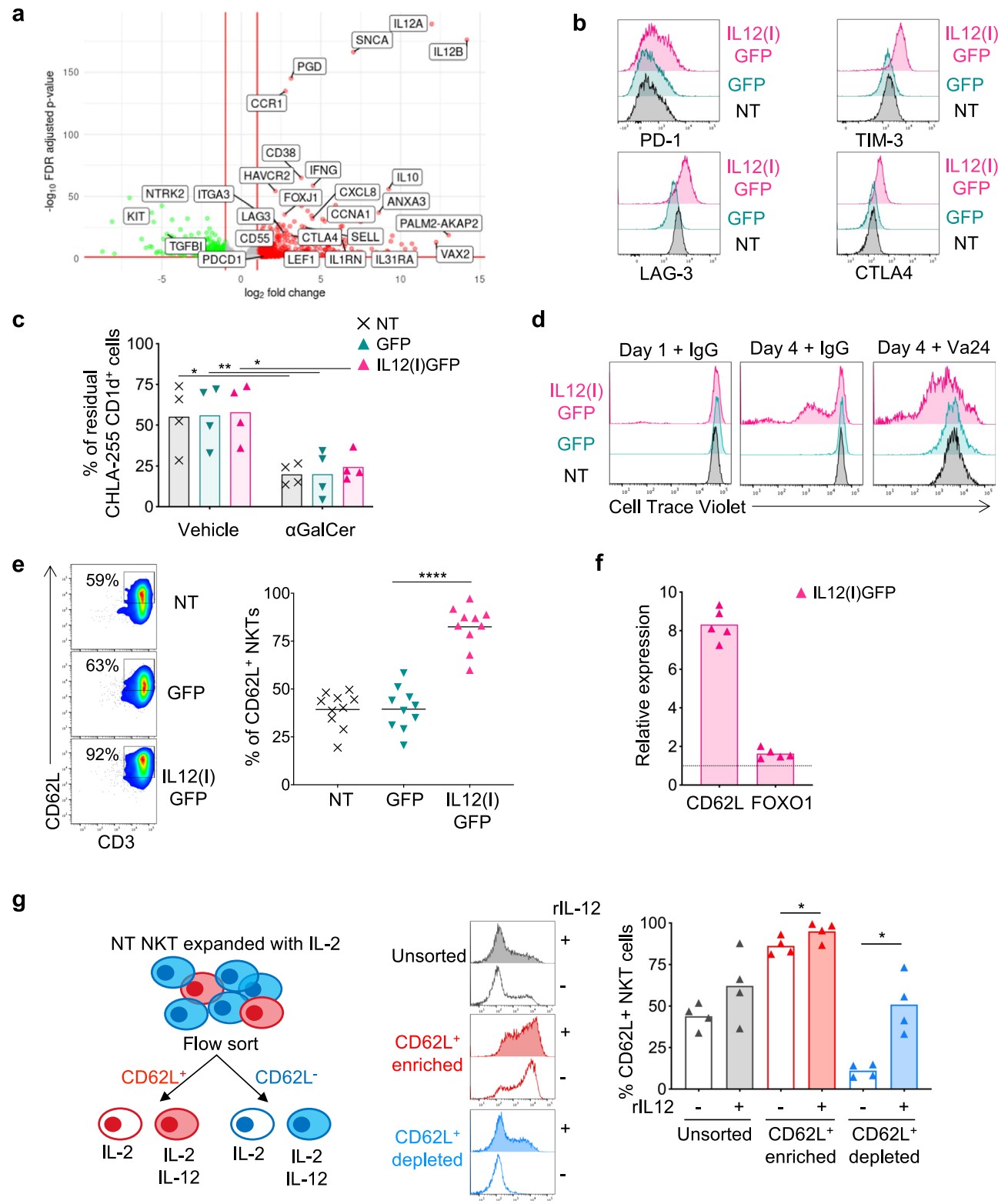

**CAR-NKTs expressing a membrane-bound form of IL-12 mediates superior antitumor effects of CAR-NKTs co-expressing IL-15**

The systemic delivery of recombinant IL-12 as well as the delivery of IL-12 by engineered tumor-infiltrating lymphocytes caused toxicity in patients[30,31]. To increase the safety profile of CAR-IL12 NKTs by promoting localized effects of IL-12, we designed a membrane-bound form of IL-12 (Fig. 7a). Specifically, we anchored IL-12 to the cell

membrane by fusing IL-12 with the human CD8α stalk. Furthermore, to prevent dimerization of the CD8α stalk of IL-12 with the CD8α stalk of the CAR, we mutated the cysteine residues located at positions 164 and 181 of the CD8α stalk region (C164S and C181S)[32]. The presence of membrane-bound IL-12 (IL12.TM) did not affect the CAR expression in NKTs (Fig. 7b and Fig. S10A). IL-12 was detectable on the cell surface only when NKTs were transduced with the membrane-bound form (Fig. 7c and Fig. S10B), and IL-12 detected in the supernatant was

**Fig. 2 | IL-12 reprograms NKTs to activated and proliferative cells. a** Volcano plot illustrating genes differentially expressed (alpha <0.2) between IL12(I)GFP and GFP NKTs. Positive LFC values reflect over-expression in the IL12(I)GFP NKTs; $n = 3$ donors. **b** Representative flow cytometry plots showing the Mean Fluorescence Intensity (MFI) of PD-1, TIM-3, LAG-3, and CTLA4 expression in NT, GFP, and IL12(I)GFP NKTs at day 14 of culture. **c** Quantification of the residual tumor cells when NKTs were cocultured with CD1d expressing CHLA-255 cells loaded with 100 ng/ml of αGalCer. Four days after the co-culture cells were collected and stained with anti-iTCR (Vβ11) and anti-GD2 Abs to identify NKTs and neuroblastoma cells, respectively, by flow cytometry. Mean is shown; $n = 4$ donors; *$p = 0.0115$ NT; **$p = 0.0098$ GFP; *$p = 0.0164$ IL12(I)GFP; two-way ANOVA. Source data are provided as a Source Data file. **d** Representative flow cytometry plots of NKTs stained with Cell Trace Violet (CTV) and stimulated with the control IgG Ab or with the iNKT Ab (Vα24).

CTV dilution was assessed on day 1 and 4; $n = 3$. **e** Representative flow cytometry plots (left panel) and summary (right panel) of the expression of CD62L in NT, GFP, and IL12(I)GFP NKTs at day 14 of culture. Mean is shown; $n = 10$ donors; ****$p < 0.0001$; two-tailed paired $t$ test. Source data are provided as a Source Data file. **f** Relative expression of CD62L and FOXO1 assessed by qPCR in IL12(I)GFP and GFP NKTs. Mean is shown; n = 5 donors. Source data are provided as a Source Data file. **g** Schematic of the NKT sorting experiment and post-sorting culture conditions (left panel) of CD62L+ and CD62L- NKTs. Representative flow cytometry plots (middle panel) and summary (right panel) of the expression of CD62L in CD62L+ and CD62L- NKTs cultured for 6 days with or without recombinant human IL-12 (10 ng/ml). Mean is shown; $n = 4$ donors; *$p = 0.0463$ in CD62L enriched; *$p = 0.0108$ in CD62L depleted; two-tailed paired $t$ test. Source data are provided as a Source Data file.

significantly reduced in GD2.CAR(I)IL12.TM NKTs compared to GD2.CAR(I)IL12 NKTs in both resting and activated conditions (Fig. S10C). IL12.TM retained functional interaction with the IL-12R complex because we observed comparable phosphorylation of STAT4 in GD2.CAR(I)IL12.TM, GD2.CAR(I)IL12 and CD19.CAR(I)IL12 NKTs (Fig. 7e). Moreover, IL12.TM continued to mediate the CD62L upregulation in GD2.CAR(I)IL12.TM NKTs (Fig. 7d and Fig. S10D). We performed in vitro coculture experiments in which GD2.CAR(I)IL12.TM NKTs and GD2.CAR(I)IL12 NKTs showed similar capacity in mediating tumor elimination (Fig. S10E), and similar Th1 polarization, but IL-12 was not detectable in the culture supernatant of GD2.CAR(I)IL12.TM NKTs (Fig. S10F). We then tested GD2.CAR(I)IL12.TM NKTs in the metastatic neuroblastoma model, and to further demonstrate the potency of the antitumor effects of NKTs expressing IL-12, we decreased the number of injected NKTs ($5 \times 10^6$ NKTs/mouse) (Fig. 7f). GD2.CAR(I)IL12 and GD2.CAR(I)IL12.TM NKTs showed similar tumor control (Fig. 7g and Fig. S10G) and increased overall survival compared to mice treated with control GD2.CAR-NKTs (Fig. 7h). Four weeks after treatment we found a significantly higher number of NKTs in mice receiving either GD2.CAR(I)IL12 or GD2.CAR(I)IL12.TM NKTs compared to GD2.CAR-NKTs (Fig. 7i). We quantified human cytokines in the plasma of the mice at sacrifice and observed that the level of IL-12 in mice treated with GD2.CAR(I)IL12.TM NKTs was comparable to those detected in GD2.CAR-NKTs treated mice, while IL-12 was detected in mice treated with GD2.CAR(I)IL12 NKTs (Fig. 7j). Confirming the Th1 polarization of GD2.CAR(I)IL12.TM NKTs, the level of IFN-γ in the plasma of mice treated with GD2.CAR(I)IL12.TM NKTs was comparable to that of mice receiving GD2.CAR(I)IL12 NKTs (Fig. 7j). Since GD2.CAR-NKTs co-expressing IL-15 have been proved to be effective in pediatric patients with neuroblastoma, we compared GD2.CAR.IL15 NKTs with GD2.CAR(I)IL12 and GD2.CAR(I)IL12.TM in the neuroblastoma murine model at the lowest dose of CAR-NKTs. Mice did not suffer from treatment-related side effects as assessed by measuring the body weight (Fig. S10H). GD2.CAR.NKTs co-expressing IL-12 showed superior antitumor effects compared to GD2.CAR co-expressing IL-15 (Fig. 7k, L and Fig. S10I). Accordingly, only GD2.CAR-NKTs co-expressing IL-12 were detected in the liver at sacrifice (Fig. S10J), supporting the unique effect of IL-12 in improving NKTs persistence in vivo. Overall, these data show that CAR-NKTs expressing a membrane-bound form of IL-12 have comparable activity of CAR-NKTs co-expressing the soluble form of IL-12, but without accumulation of IL-12 in the plasma of treated mice. CAR-NKTs co-expressing IL-12 also outperform the clinically effective CAR-NKTs co-expressing IL-15 suggesting that IL-12 may further improve the therapeutic activity of CAR-NKTs.

## Discussion

We report that IL-12 expression in human NKTs and CAR-NKTs promotes an extensive molecular reprogramming that leads to the generation of Th1-polarized cells with memory features and extended persistence and antitumor capability.

Cells of the innate immune system such as NKs and NKTs expressing CARs reached the clinical application with promising outcomes[3,15]. It is also evident from these clinical studies that the co-expression of survival factors such as IL-15 is critical in supporting the expansion and persistence of both CAR-NKs and CAR-NKTs in patients[3,15,16]. The beneficial effects of adding IL-15 to both CAR-NKs and CAR-NKTs were supported by preclinical experiments conducted in xenotransplant models in immunodeficient mice[2,33]. Here we report that a pro-inflammatory cytokine such as IL-12 plays a previously non-appreciated role in promoting memory formation, survival and proliferative capacities in human NKTs.

Our results demonstrate that human NKTs can be engineered to express IL-12 without affecting their overall purity. IL-12 released by NKTs engages the IL-12 receptor expressed by NKTs, activates the IL-12 receptor pathway via STAT4 phosphorylation, and leads to the transcriptional initiation of a pro-inflammatory program. In particular, IL-12, but not other cytokines such as IL-15 or IL-21[28] generates Th1 polarized polyfunctional NKTs releasing more than two pro-inflammatory cytokines, while the release of inhibitory cytokines such as IL-4 is attenuated. The generation of polyfunctional CAR-T cells has been correlated with superior antitumor activity in clinical studies[34,35], and IL-12 seems to reprogram NKTs to polyfunctional Th1 polarized cells.

In addition to inflammation, IL-12 is involved in CD8 T-cell differentiation in response to pathogens. Specifically, it has been shown that both IL-12 and type I interferon transiently expressed in response to pathogens participate directly in developing CD8 memory cells characterized by high expression of CD62L[36]. We have identified CD62L expression as a marker of memory NKTs[17], and CD62L expression correlates with a subset of CAR-NKTs showing superior antitumor activity in patients[15]. Here we show that IL-12 causes upregulation of CD62L in CD62L-negative human NKTs in vitro indicating that IL-12 maintains the same mode of action in CD8 T cells and innate NKTs in promoting CD62L expression and memory differentiation. Furthermore, this evidence indicates that innate NKTs circulating in the peripheral blood in humans possess an intrinsic differentiation hierarchy and can differentiate into memory cells in response to IL-12. However, our data also reveal that while CD62L is clearly a marker of memory NKTs, CD62L may not play a critical role in supporting long-term proliferation and persistence in vivo in immune deficient mice. We found that NKTs expressing IL-12 proliferate and persist long-term in vivo despite *SELL* KO, while overexpression of CD62L in the absence of IL-12 does not enhance NKT in vivo persistence. In contrast, we found that IL-12 causes the upregulation of the Wnt/β-catenin transcription factor lymphoid enhancer binding factor 1 (LEF1), which we recently reported being involved in the formation of memory CD62L-expressing NKTs[26]. Thus IL-12 causes an extensive reprogramming of human NKTs, which includes the formation of long-lived memory cells expressing CD62L and LEF1.

Chronic inflammation and chronic exposure to inflammatory cytokines are generally considered having negative effects on immune

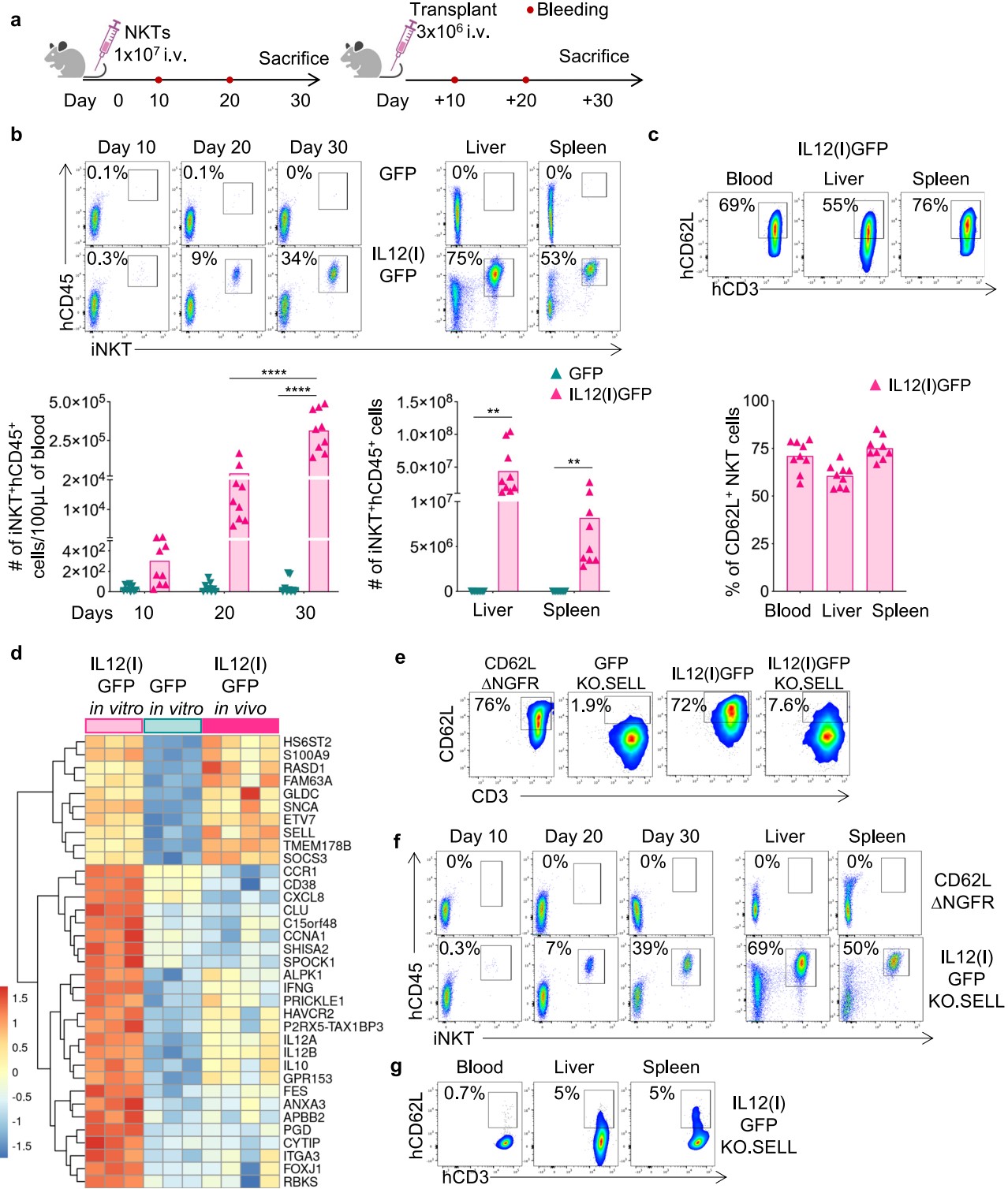

responses against both pathogens and tumors. Here we found an unexpected positive function of IL-12 constitutively expressed in human NKTs. We observed that the continuous exposure of NKTs to IL-12 not only maintains their Th1 polarization, but is required to support NKT cell proliferation and long-term in vivo persistence in re-transplant models upon adoptive transfer. Furthermore, long-term in vivo persisting IL-12-expressing NKTs maintained a molecular signature very similar to that observed in vitro, which is characterized by high expression of effector molecules and inhibitory receptors such as PD1, LAG3, and TIM3, but simultaneous expression of memory markers

such as CD62L and LEF1. Thus IL-12 Th1 polarized NKTs display long-term persistence capacity rather than exhausted characteristics, and mirror the molecular signature observed in long-term persisting CD4[+] CD19-specific CAR-T cells expressing granzymes and PD1 as well as memory markers such as CCR7[37].

We observed that human NKTs co-expressing a CAR and IL-12 retain the functionality of both molecules, and IL-12 generates Th1 polarized CAR-NKTs with enhanced proliferative and persistence capacities, which led to improved antitumor activity. IL-12 was superior to IL-15 in promoting the antitumor effects of CAR-NKTs suggesting

**Fig. 3 | NKTs expressing IL-12 acquire long-term longevity in vivo. a** Schematic representation of the in vivo experiment in which non-tumor bearing NSG mice are infused with either GFP or IL12(I)GFP NKTs. **b** Representative flow cytometry plots (upper panel) and summary (lower panel) of the quantification of human NKTs (iNKT⁺CD45⁺) in peripheral blood samples collected 10, 20, and 30 days after GFP or IL12(i)GFP NKT infusion, and in livers and spleens at the time of euthanasia. Mean is shown; $n = 9$ mice, from two independent experiments; ****$p < 0.0001$; **$p = 0.0023$ in livers; **$p = 0.0059$ in spleens; two-way ANOVA with Bonferroni correction. Source data are provided as a Source Data file. **c** Representative flow cytometry plots (upper panel) and summary (lower panel) of the expression of CD62L in IL12(I)GFP NKTs at the time of euthanasia in the peripheral blood, livers and spleens. Mean is shown; $n = 9$ mice. Source data are provided as a Source Data file.

**d** Genes differentially expressed in IL12(I)GFP and GFP NKTs expanded in vitro, and in IL12(I)GFP collected in vivo at the time of euthanasia (FDR < 0.05, top 35 genes); $n = 3$ donors for in vitro and $n = 4$ mice for in vivo. **e** Representative flow cytometry plots of 3 donors showing the expression of CD62L in NKTs overexpressing CD62L (CD62L.ΔNGFR), expressing GFP with the knockout of CD62L (GFP KO.SELL) and in IL12(I)GFP and IL12(I)GFP NKTs with the knockout of CD62L (IL12(I)GFP KO.SELL) at day 14 of culture. **f** Representative flow cytometry plots of 4 mice showing the quantification of human NKTs (iNKT⁺CD45⁺) in peripheral blood samples collected 10, 20 and 30 days after IL12(i)GFP KO.SELL or CD62L.ΔNGFR NKTs infusion, and in livers and spleens at the time of euthanasia. **g** Representative flow cytometry plots of 4 mice showing the expression of CD62L in IL12(I)GFP KO.SELL NKTs at the sacrifice in peripheral blood, livers and spleens.

that this cytokine may further increase the therapeutic effects of CAR-NKTs. Of note, IL-12 expression was sufficient in maintaining proliferation and expansion of CAR-NKTs in the absence of specific antigen stimulation through the CAR. In both leukemia and neuroblastoma models, we observed that CAR-NKTs expressing a non-tumor specific CAR, but encoding IL-12, persisted in tumor-bearing mice, but did not control the tumor indicating that IL12 exploits his pro-survival function in NKTs in the absence of CAR engagement. Furthermore, RNAseq data of CAR-NKTs expressing IL-12 and clearing the tumor long-term in vivo showed a molecular signature very similar to NKTs expressing IL-12 only. This suggests that IL-12 signaling defines the molecular signature of NKTs co-expressing both CAR and IL-12.

Administration of recombinant IL-12 caused toxicity in humans and multiple efforts have been devised to develop local delivery of IL-12 within the tumor. Expression of IL-12 under the control of an inducible promoter in TILs was insufficient in mitigating IL-12-associated toxicities in patients[31]. Secreted cytokines including IL-12 can be anchored on the surface of mammalian cells promoting local retention of bioactive cytokines[38,39]. Membrane-bound cytokines can be generated using either suitable transmembrane domains or GPI-anchored sequences. However, GPI-anchored proteins can be shed or cleaved making them soluble[40]. Here we show that the CD8α stalk including mutations to prevent the formation of disulfide bonds[32] allows functional expression of membrane-bound IL-12 in NKTs without interfering with the CAR function and preventing the accumulation of IL-12 in the plasma of treated mice.

Overall, we showed that IL-12 reprograms human NKTs to long-lived Th1-polarized cells with sustained antitumor activity. Taking into consideration the promising activity of IL-15-expressing CAR-NKTs in patients with relapsed/refractory neuroblastoma, IL-12 engineering could represent the next step to manufacture long-term persisting innate cells for cancer immunotherapy in solid tumors.

## Methods

### Study design
This study was designed to enhance CAR-NKT anti-tumor functions by the incorporation of IL-12. To validate the efficacy of CAR-NKTs, in vitro and in vivo functional assays were performed using a variety of hematological and solid-established tumor cell lines. NKT cells were isolated from commercially available PBMC obtained from healthy donors through Gulf Coast Blood Bank. The Institutional Animal Care and Use Committee (IACUC) approved protocols for all animal work.

### Cell lines
B cell leukemia cell line BV-173 was purchased from DSMZ and Burkitt's lymphoma cell line Daudi was purchased from ATCC. Human neuroblastoma cell lines CHLA-255 and firefly luciferase-labeled (FFLuc)-CHLA-255 were gifts from Dr. Leonid Metelitsa at Baylor College of Medicine (originally derived from a metastatic lesion in the brain in a patient with recurrent disease at Children's Hospital Los Angeles) and LAN-1 was a gift from Dr. Malcom Brenner at Baylor College of Medicine, originally purchased from ATCC. Cell lines were maintained in

culture with RPMI 1640 (Gibco) supplemented with 10% FBS (Sigma), 1% L-glutamine (Gibco), and 1% penicillin-streptomycin (Gibco) in a humidified atmosphere containing 5% CO2 at 37 °C. In selected experiments, the CHLA-255 cell line was transduced with an SFG gamma retroviral vector encoding the CD1d gene. In selected experiments, the BV-173 and CHLA-255 cell lines were transduced to express the fusion protein firefly luciferase and enhanced GFP (eGFP-FFluc). Cells were kept in culture for less than 2 consecutive months, after which aliquots from the original expanded vial were used. All tumor cell lines were routinely tested to exclude contamination with mycoplasma and assessed for the expression of tumor markers by flow cytometry to confirm identity.

### Retroviral constructs and generation of retroviral supernatants
All retroviral vectors were generated using the SFG backbone. The sequence of the p40 and p35 subunits of IL-12 were obtained from the NCBI website (NM_002187.2 and NM_000882.2 respectively) and were linked together with a flexible linker described from Anderson R. et al.[21] through overlapping PCR. The whole IL-12 cassette was cloned into the SFG retroviral vector containing the IRES and GFP or ΔNGFR. As control vectors containing the IRES and GFP or ΔNGFR only were used. The sequence of IL-21 was obtained from the NCBI website (NM_021803.3) and it was cloned into the SFG retroviral vector. Vectors encoding IL-15, GD2.CAR, GD2.CAR.IL15 and CD19.CAR were previously described[25,29,41,42]. IL-12 was then cloned in the CAR cassettes separated by the IRES element (GD2.CAR(I)IL12, CSPG4.CAR(I)IL12 and CD19.CAR(I)IL12). The sequence of CD62L (SELL) was obtained from NCBI (NM_000655) and made by gene synthesis (Genewiz, Azenta Life Sciences) and cloned in the SFG vector containing the IRES and the ΔNGFR as selection marker. The sequence of CD1d (CD1D) was obtained from NCBI (NM_001766.3) and made by gene synthesis (Genewiz, Azenta Life Sciences) and cloned in the SFG vector. To generate the membrane-bound IL-12 we added the CD8α stalk with the mutations C164S and C181S to prevent dimerization with the CAR[32]. Retroviral supernatants were prepared by transient transfection of 293 T cells.

### NKT cell isolation, transduction and in vitro expansion
Buffy coats from healthy volunteer blood donors were purchased from the vendor Gulf Coast Regional Blood Center (Houston, Texas, USA). The protocol for the use of purchased buffy coat has been submitted to the institutional IRB and it is considered IRB exempt because it uses deidentified commercial products. Peripheral blood mononuclear cells (PBMCs) were isolated by Lymphoprep (Accurate Chemical and Scientific Corporation) density gradient centrifugation. NKTs were purified from PBMCs as previously described[20]. Briefly, NKTs they were activated with autologous irradiated PBMCs (40 Gy, RS-2000 Biological System) loaded with α-Galactosylceramide (α-GalCer, 100 ng/ml, Diagnocine LLC) at a PBMC:NKT ratio 10:1 in presence of IL-2 (200 IU/ml, R&D). NKTs were transduced in retronectin coated plates (Takara, 7 μg/ml) on day 5 and further expanded for 10 days in the presence of IL-2 and then used for functional assays.

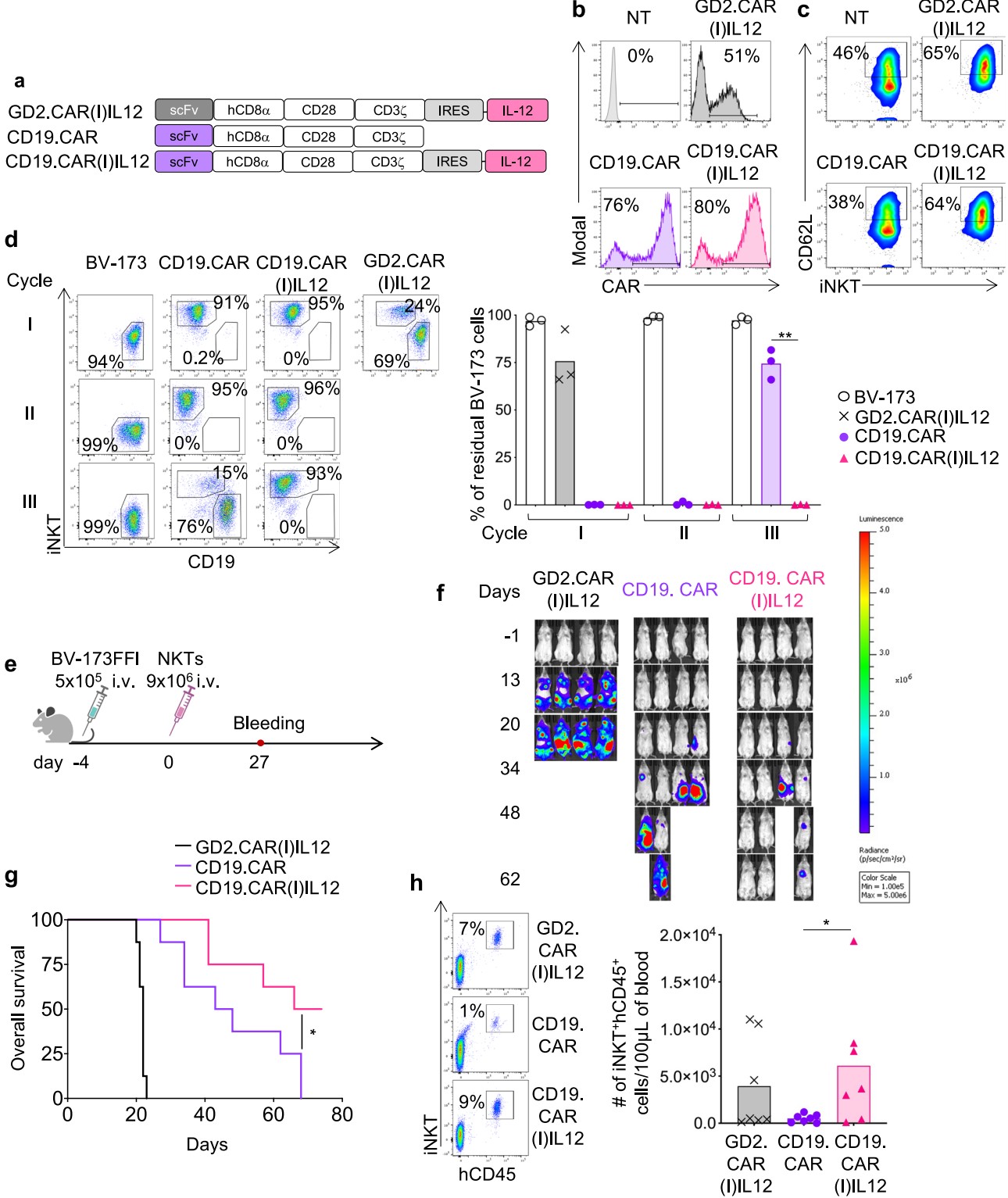

**Fig. 4 | CD19.CAR(I)IL12 NKTs are superior to CD19.CAR-NKTs in eliminating tumor cells. a** Schematic of the retroviral vectors encoding the CD19.CAR or GD2.CAR and IL-12 used to transduce human NKTs. **b** Representative flow cytometry plots of 5 donors showing CAR expression in control (NT) and CAR-NKTs assessed at day 14 of culture. **c** Representative flow cytometry plots of 5 donors showing the CD62L expression in NT and CAR-NKTs assessed at day 14 of culture. **d** Representative flow cytometry plots (left) and summary (right) of the quantification of residual tumor cells after each cycle when CAR-NKTs were co-cultured with BV-173 tumor cells (E:T = 1:2). Collected cells were stained with the iNKT Ab and scFv-specific CD19 Ab to identify NKTs and leukemia cells, respectively, by flow cytometry. Mean is shown; n = 3 donors; **p = 0.0039, two-tailed paired t test.

Source data are provided as a Source Data file. **e** Schematic representation of the xenograft leukemia model in NSG mice. Mice were engrafted i.v. with 5 × 105 BV-173 Firefly-luciferase labeled tumor cells. Four days later, mice received i.v. 9 × 10⁶ CD19.CAR-NKTs. **f** Representative tumor BLI. **g** Kaplan–Meier survival curve showing overall survival; n = 7 mice; p = 0.0438; log-rank (Mantel-Cox) test. Source data are provided as a Source Data file. **h** Representative flow cytometry plots (left panels) and quantification (right panel) of human NKTs (iNKT+hCD45+) in peripheral blood samples collected 4 weeks after NKT infusion. Mean is shown; n = 7 mice; *p = 0.0470, two-tailed unpaired t test. Source data are provided as a Source Data file.

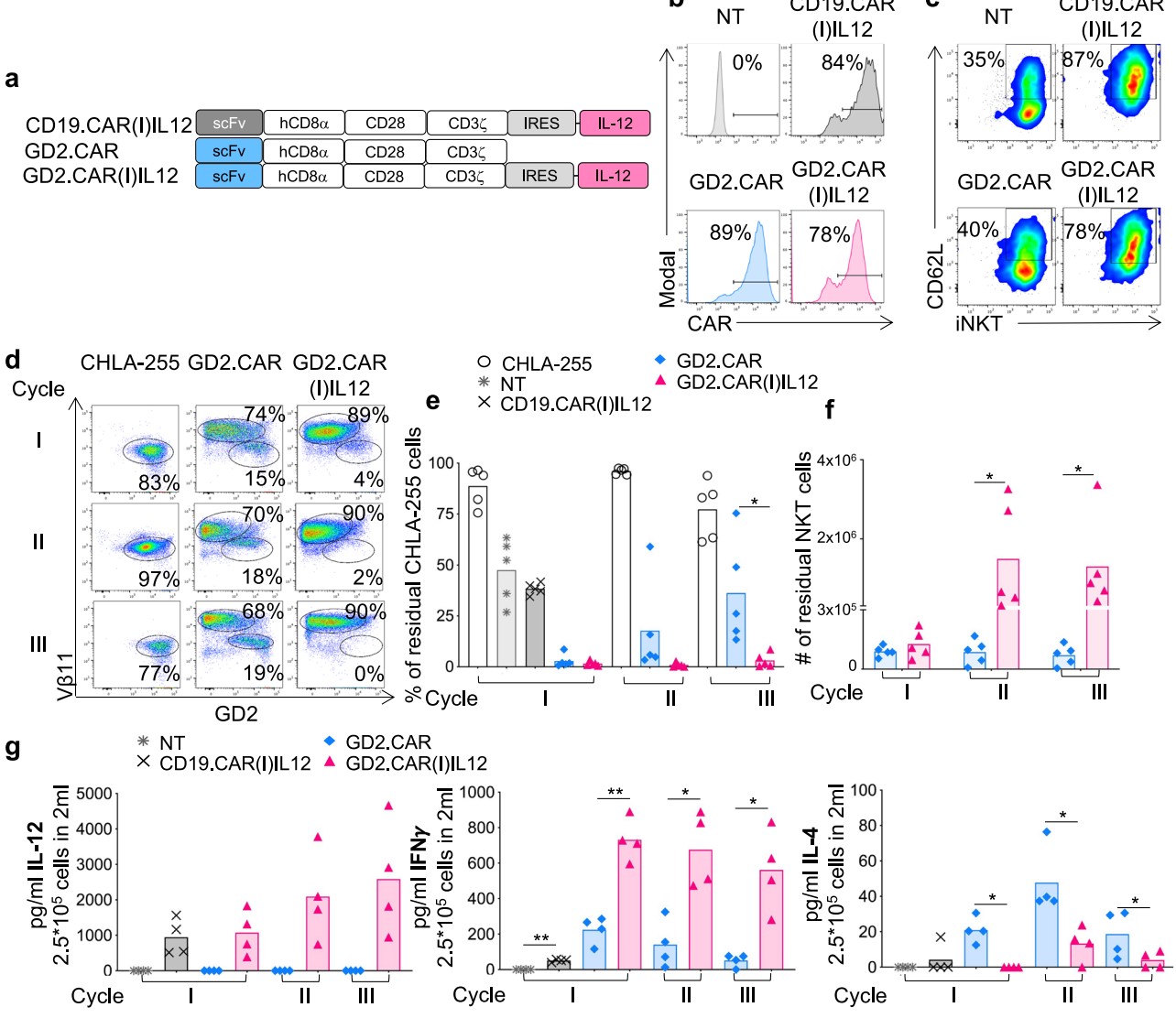

**Fig. 5 | GD2.CAR(I)IL12 NKTs are superior to GD2.CAR-NKTs in eliminating tumor cells in stress conditions in vitro and acquire a pro-inflammatory phenotype. a** Schematic of the retroviral vectors encoding the CD19.CAR or GD2.CAR and IL-12 used to transduce NKTs. **b** Representative flow cytometry plots of 8 donors showing the CAR expression in control (NT) and CAR-NKTs assessed at day 14 of culture. **c** Representative flow cytometry plots of 8 donors showing the CD62L expression in NT and CAR-NKTs assessed at day 14 of culture. **d–f** Representative flow cytometry plots (**d**), and summary of the quantification of residual tumor cells (**e**) and NKTs (**f**) after each cycle when NKTs were co-cultured with CHLA-255 (E:T = 1:1). Cells were collected and stained with anti-iTCR (Vβ11) and anti-GD2 Abs

to identify NKTs and neuroblastoma cells, respectively, by flow cytometry. Mean is shown; $n = 5$ donors; *$p = 0.0458$ cycle III, % tumor cells; # of NKTs; $p = 0.0457$ cycle II and III; two-way ANOVA. Source data are provided as a Source Data file. **g** Quantification of IL-12, IFN-γ and IL-4 produced by NT, CD19.CAR(I)IL12, GD2.CAR and GD2.CAR(I)IL12 NKTs when cocultured with CHLA-255 at E:T ratio 1:1. Cytokines were measured in supernatants collected 24 h after plating $2.5 \times 10^5$ NKT cells/well with $2.5 \times 10^5$ CHLA-255/well in 24 well plate in 2 mL of complete media without cytokines. Mean is shown; $n = 4$ donors; IFN **$p = 0.0010$, **$p = 0.0037$, *$p = 0.0387$, *$p = 0.0239$; IL-4, *$p = 0.0110$, * $p = 0.0152$, *$p = 0.0438$; two-tailed paired $t$ test. Source data are provided as a Source Data file.

## Immunophenotyping

NKTs were stained with antibodies (Ab) against CD3 (APC-H7, clone SK7), CD62L (BV605, clone DREG-56), CD4 (PE-Cy7, clone SK3), CD8 (Alexa Fluor 700, clone RPA-T8), CD19 (APC, clone HIB19), CD45 (APC, clone 2D1), CD271 (NGFR, APC, clone C40-1457), CD279 (PD-1, PE-Cy7, clone EH12.1), CD366 (TIM-3, BV711, clone 7D3), CD223 (LAG-3, PE, clone T47-530) and CD152 (CTLA-4, BV421, clone BNI3) from BD Biosciences; IL-12 (p70, APC) and CD212 (IL12R b2, APC) from Miltenyi Biotech. Tumor cells were stained with Abs against GD2 (PE, clone 14.G2a) and CD276 (B7-H3, BV421, clone 7-517) from BD Biosciences. The purity of NKTs was assessed by staining the cells with the PE-conjugated Abs specific for TCR Vα24 chain (iNKT, clone 6B11, BD Biosciences) or for TCR β11 chain (FITC, Beckman Coulter), which we have previously

shown to be superimposable[20]. The expression of the CD19.CAR, GD2.CAR and CSPG4.CAR was assessed using specific anti-idyotipic Abs, followed by the staining with a secondary goat anti-Mouse Ab (APC, Ig multiple adsorption, BD Biosciences). Data acquisition was performed on a BD FACSCanto II or BD LSRFortessa using the BD FACS-Diva software. Data analyses were performed with the FlowJo software (BD Biosciences). To detect apoptosis, NKTs ($5 \times 10^5$ cells/well) were activated with the iNKT Ab (5 μl in 1 ml of PBS), and 48 h later NKTs were collected and stained using Annexin-V/7AAD staining (BD Biosciences) according to the manufacture's protocol. Data acquisition was performed on a BD LSRFortessa using the BD FACS-Diva software. Data analyses were performed with the FlowJo software. The quantity of antibodies used was according to manufactures suggestions.

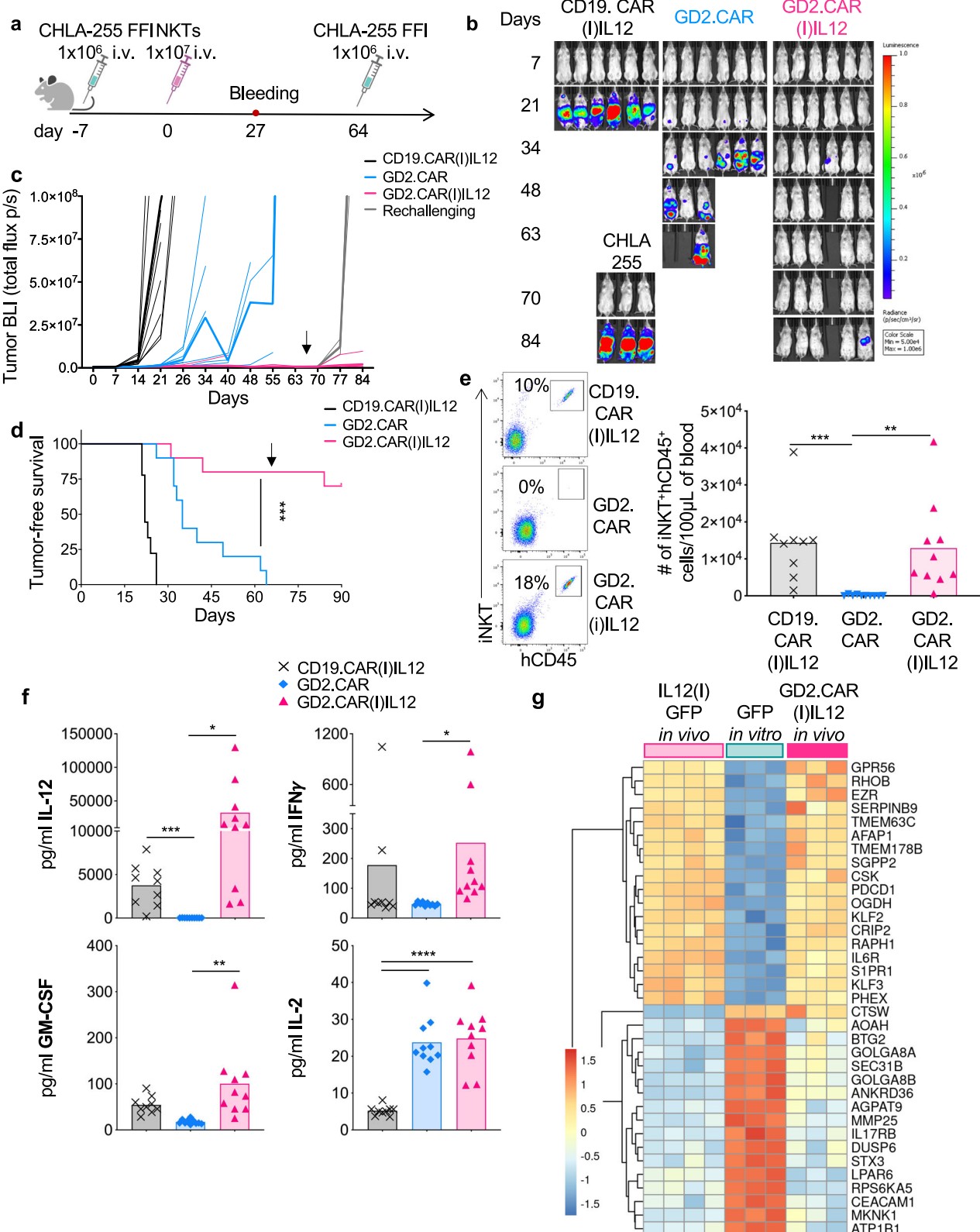

## Western Blot

Protein lysate was resolved on 4% -15% SDS polyacrylamide gel electrophoresis gels (SDS-PAGE, Bio-Rad). After protein transfer onto Polyvinylidene fluoride membranes (Bio-Rad), membranes were blocked in 5% non-fat milk in TBS-T and incubated with primary and secondary Abs in TBS-T with 1% milk. The following Abs were used: α-Stat4 (C46B10, dilution 1:1000) and α-Phospho-Stat4 (Tyr693, dilution 1:1000) from Cell Signaling; α-CD3ζ (dilution 1:1000) horseradish peroxidase (HRP) conjugated from Santa Cruz; HRP conjugated secondary Ab(Goat-α-Rabbit #32460, dilution 1:500) from Thermo Scientific. Incubation with the primary Abs was performed overnight at 4 °C while the incubation with the secondary Ab was performed for 1 h

**Fig. 6 | Transgenic IL-12 promotes antitumor activity and persistence of GD2.CAR(I)IL12 NKTs in a re-challenge neuroblastoma tumor model.**
**a** Schematic representation of tumor re-challenge in the metastatic xenograft neuroblastoma model. Mice were engrafted i.v. with $1 \times 10^6$ CHLA-255 Firefly-luciferase labeled neuroblastoma tumor cells. Seven days later mice received i.v. $1 \times 10^7$ CAR-NKTs. Mice that were tumor free at day 64, received a second i.v injection of $1 \times 10^6$ CHLA-255 cells. A group of mice infused with CHLA-255 cells only was used as a second control for this model. **b**, **c** Representative tumor BLI (**b**) and BLI tumor kinetics (**c**). Thin lines represent individual mice, and bold lines represent the mean for the group; n = 9 mice for CD19.CAR(I)IL12 and $n = 10$ mice for GD2.CAR or GD2.CAR(I)IL12. Source data are provided as a Source Data file. **d** Kaplan–Meier survival curve showing tumor-free survival; $n = 9$ mice for CD19.CAR(I)IL12 and

$n = 10$ mice for GD2.CAR or GD2.CAR(I)IL12, ***$p = 0.0004$; log-rank Mantel-Cox test. Source data are provided as a Source Data file. **e** Quantification of human NKTs (iNKT$^+$CD45$^+$) in peripheral blood samples collected 4 weeks after NKT infusion. Mean is shown; $n = 9$ mice for CD19.CAR(I)IL12 and $n = 10$ mice for GD2.CAR or GD2.CAR(I)IL12; **$p = 0.0038$, ***$p = 0.0005$, two-tailed unpaired $t$ test. Source data are provided as a Source Data file. **f** Quantification of human IL-12, IFN-γ, GM-CSF and IL-2 in the serum of mice 4 weeks after the injection of CD19.CAR(I)IL12, GD2.CAR or GD2.CAR(I)IL12 NKTs. Mean is shown; $n = 10$ mice; IL-12 ***$p = 0.0001$, *$p = 0.0250$; IFN-γ *$p = 0.0453$; GM-CSF **$p = 0.0054$; IL-2 ****$p < 0.0001$; two-tailed paired t test. Source data are provided as a Source Data file. G. Genes differentially expressed in CAR.GD2(I)IL12 NKTs collected at the time of euthanasia versus IL12(I) GFP NKTs collected in vivo at day 30.

at R.T. Membranes were developed either with Clarity Max Western ECL Substarte (Bio-Rad) or with SuperSignal West Femto Maximum Sensitivity Substrate (Thermo Scientific) on a Gel station (Bio-Rad).

### Real-time qPCR
Gene expression was measured in NKTs by quantitative polymerase chain reaction (qPCR) with specific primers and probes and normalized to the 18 S rRNA gene expression. Briefly, RNA was extracted from NKTs (RNeasy Plus Kit, Qiagen) and 1 μg of RNA was used for reverse transcription (5X VILO Reaction Mix and 10X SuperScript Enzyme Mix from Invitrogen). For the qPCR reaction (TaqMan 2X Universal PCR Master Mix from Life Technologies) 20 ng of cDNA were used in duplicates. The relative expression was calculated as follows: 2^-[(CTgene − CT18S) − CTVα24 in NT-NKTs]. Data acquisition was performed on a QuantStudio 6 Flex from Life Technologies using the QuantStudio Real-Time PCR software. The primers and probe for the 18S rRNA (Hs03003631_g1), CD62L (Hs00174151_m1), IL12β (Hs01011518_m1), IFNγ (Hs00989291_m1), FOXO1 (Hs00231106_m1) and TBX21 (T-bet, Hs00203436_m1) were purchased from Thermo Fisher Scientific.

### IsoPlexis IsoCode secretome
NKTs ($5 \times 10^5$ cells/well) were cultured in 24-well plates coated with the iNKT Ab (5 μl in 1 ml of PBS) or with IgG Ab (5 μl in 1 ml of PBS) as negative control for 18 h. NKTs were labeled with membrane stain (1:500 dilution, IsoPlexis) for on chip cell detection and resuspended at a density of $1 \times 10^6$ cells/mL. Approximately 30 microliters of cell suspension, equivalent to 30,000 cells, was loaded into a human adaptive IsoCode Chip (IsoPlexis). This chip comprised around 12,000 cellular microchambers, each pre-patterned with a complete copy of the 32-plex antibody array for the assessment of single-cell secretomics. The cells on the chip underwent a 13.5-h incubation at 37 °C with 5% CO2 using the IsoLight automation system by IsoPlexis. After this final incubation, proteins secreted by approximately 1000 single cells were captured by the 32-plex antibody barcoded chip and subjected to analysis through a backend fluorescence ELISA-based assay. Polyfunctionality of T cells, defined as cells co-secreting 2 or more cytokines, was examined using the IsoSpeak software across five functional groups: Effector (Granzyme B, TNF-α, IFN-γ, MIP1-α, Perforin, TNF-β); Stimulatory (GM-CSF, IL-2, IL-5, IL-7, IL-8, IL-9, IL-12, IL-15, IL-21); Chemoattractive (CCL11, IP-10, MIP-1β, RANTES); Regulatory (IL-4, IL-10, IL-13, IL-22, sCD137, sCD40L, TGF-β1); Inflammatory (IL-6, IL-17A, IL-17F, MCP-1, MCP-4, IL-1β). Protein signals from microchambers with no cells were used to establish cytokine-specific background. Cutoffs for any given cytokine were calculated based on background levels from wells lacking cells plus 3 standard deviations. Additionally, signals with a signal-to-noise ratio (SNR) of at least 2 (relative to the background threshold) and originating from at least 20 single cells or 2% of all single cells (whichever quantity was larger) were deemed significantly secreted.

The polyfunctional strength Index (PSI) of T cells was computed using a pre-specified formula, defined as the percentage of polyfunctional cells, multiplied by the sum of the mean fluorescence intensity (MFI) of the proteins secreted by those cells[43].

$$PSI_{sample} = (\%polyfunctional\ cells\ in\ sample) \sum_{i=1}^{32} MFI\ of\ secreted\ protein\ i\ of\ the\ polyfunctional\ cells$$

PAT-PCA is a linear dimensionality reduction method to explain the maximum variance as a simple linear combination of cytokine expression. The principal components are labeled according to their correlation with specific cytokines. Each dot represented a single cell. Similarly for the hierarchical clustering input, PCA is applied on a binarized dataset (0 = no secretion, 1 = secretion), to focus on visualizing combinatorial differences, rather than intensity differences. Circles representing the same functional group are randomly offset but remain within a radius proportional to the secretion frequency of the corresponding group (i.e., large groups = large circles, small groups = small circles). Plotting the polyfunctional subsets in such a manner allows overall similarities and differences in the donor profiles to emerge.

### NKT electroporation
The CRISPR-Cas9 system was used to knock out CD62L in NKTs. Briefly, 500 μM of sgRNA for *SELL* (ACACCTGCAACTGTGATGTG) were incubated with 20 μg of S.p. Cas9 Nuclease V3 (IDT) for 15 min at room temperature. $2 \times 10^6$ NKTs were added to the sgRNA-Cas9 mix in a final volume of 100 μl of Buffer R (Thermo Fisher). NKTs were electroporated at day 3 post activation with the Neon Transfection System (Thermo Fisher).

### Repetitive coculture
NKTs ($1.25 \times 10^5$ cells/well) were co-cultured with BV-173 or Daudi at an E:T ratio of 1:2 in 24-well plates in the absence of cytokines (Cycle I). At day 4 all cells were transferred into a new well pre-seeded with $2.5 \times 10^5$ tumor cells (Cycle II). At day 3 all cells were transferred into a new well pre-seeded with $5 \times 10^5$ tumor cells (Cycle III). At the end of every cycle cells were harvested and stained for iNKT and CD19 mAbs to detect NKTs and tumor cells, respectively. NKTs ($2 \times 10^5$ cells/well) were co-cultured with CHLA-255 or LAN-1 at an E:T ratio of 1:1 in 24-well plates in the absence of cytokines (Cycle I). On day 2 all cells were transferred into a new well pre-seeded with $2 \times 10^5$ NB cells (Cycle II). On day 4 all cells were transferred into a new well pre-seeded with $2 \times 10^5$ NB cells (Cycle III). At the end of every cycle cells were harvested and stained for CD3 or Vβ11 and GD2 mAbs to detect NKTs and tumor cells, respectively. The number of residual tumor cells in culture were enumerated by flow cytometry using CountBright absolute counting beads (Invitrogen). Data acquisition was performed on a BD FACS-Canto II or BD LSRFortessa using the BD FACS-Diva software. Data analyses were performed with the FlowJo software.

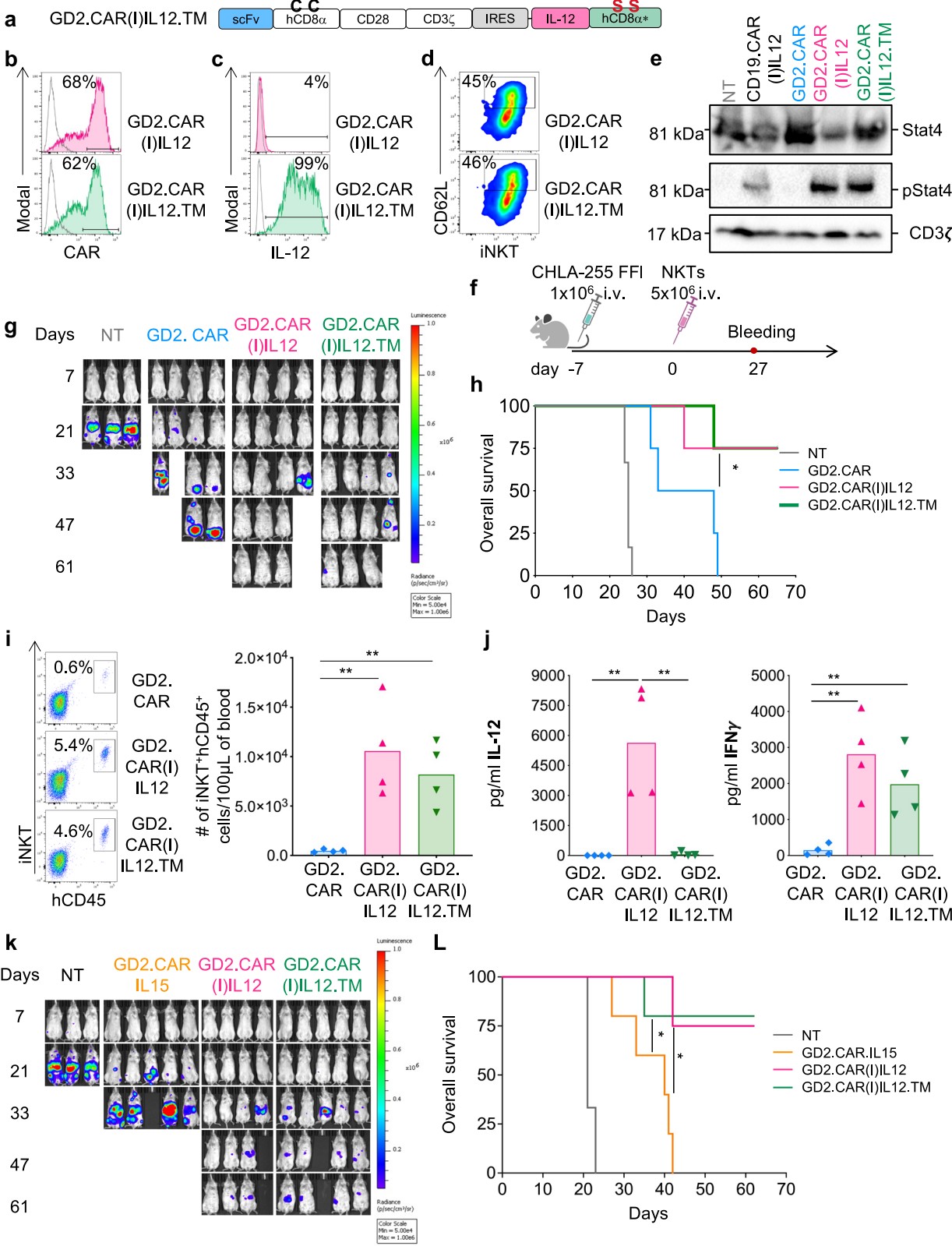

## Coculture

NKTs ($1 \times 10^5$ cells/well) were co-cultured with CHLA-255 at an E:T ratio of 1:2 in 24-well plates in the absence of cytokines. After 4 days the cells were harvested and stained for iNKT and CD276 mAbs to detect NKTs and tumor cells, respectively. Data acquisition was performed on a BD FACSCanto II or BD LSRFortessa using the BD FACS-Diva software. Data analyses were performed with the FlowJo software.

## ELISA

NKTs ($5 \times 105$ cells/well) were cultured in 24-well plates coated with the iNKT Ab (5 µl in 1 ml of PBS) or with the IgG Ab as negative control and CD28 Ab (1 mg/mL, BD Biosciences). Wells non-coated were used as negative control. Supernatants were harvested after 24 h of culture and IFN-γ, IL-4, IL-12, IL-15 and IL-21 measured using a specific ELISA (R&D System). Supernatants were also harvested after 24 h of culture

**Fig. 7 | Membrane-bound IL-12 and soluble IL-12 have comparable effects in NKTs and show superior antitumor effects compared to IL-15-expressing NKTs. a** Schematic of the retroviral vector encoding the GD2.CAR and membrane-bound IL-12 in which the mutated CD8α stalk was added to IL-12 (IL12M). **b** Representative flow cytometry plots of 5 donors showing CAR expression in GD2.CAR(I)IL12 and GD2.CAR(I)IL12.TM NKTs assessed at day 14 of culture. Gray line represents non-transduced NKTs. **c** Representative flow cytometry plots of 5 donors showing the expression of IL-12 on the cell surface of GD2.CAR(I)IL12 and GD2.CAR(I)IL12.TM NKTs assessed at day 14 of culture. Gray line represents non-transduced NKTs. **d** Representative flow cytometry plots of 5 donors showing CD62L expression in GD2.CAR(I)IL12 and GD2.CAR(I)IL12.TM NKTs assessed at day 14 of culture. **e** Representative western blot illustrating STAT4 phosphorylation in NT, CD19.CAR(I)IL12, GD2.CAR, GD2.CAR(I)IL12 and GD2.CAR(I)IL12.TM NKTs at day 14 of culture; n = 3. Source data are provided as a Source Data file. **f** Schematic representation of the metastatic xenograft neuroblastoma model. Mice were engrafted i.v. with $1 \times 10^6$ CHLA-255 Firefly-luciferase labeled neuroblastoma tumor cells and treated with 5 × 106 NT, GD2.CAR, GD2.CAR(I)IL12 and GD2.CAR(I)IL12.TM NKTs. **g** Representative tumor BLI; n = 4 from two independent experiments. **h** Kaplan−Meier survival curve showing overall survival; n = 4 mice, *p = 0.0344 GD2.CAR vs GD2.CAR(I)IL12.TM NKTs; log-rank Mantel−Cox test. Source data are provided as a Source Data file. **i** Representative flow cytometry plots (left panels) and quantification of human NKTs (iNKT⁺CD45⁺) in peripheral blood samples collected 4 weeks after NKT infusion. Mean is shown; n = 4 mice; **p = 0.0060 GD2.CAR vs GD2.CAR(I)IL12 NKTs, **p = 0.0035 GD2.CAR vs GD2.CAR(I)IL12.TM NKTs, two-tailed unpaired t test. Source data are provided as a Source Data file. **j** Quantification of IL-12 and IFN-γ detected in the serum of mice at sacrifice. Mean is shown; n = 4 mice; IL-12: **p = 0.0077 GD2.CAR vs GD2.CAR(I)IL12, **p = 0.0081 GD2.CAR(I)IL12 vs GD2.CAR(I)IL12.TM; IFNγ: **p = 0.0032 GD2.CAR vs GD2.CAR(I)IL12, **p = 0.0086 GD2.CAR vs GD2.CAR(I)IL12.TM; two-tailed paired t test. Source data are provided as a Source Data file. **k, L** Metastatic xenograft neuroblastoma model in which mice were treated with GD2.CAR.IL15 or GD2.CAR(I)IL12 or GD2.CAR(I)IL12.TM NKTs following the tumor model described in (**f**). **k** Representative tumor BLI. **L** Kaplan−Meier survival curve showing overall survival; n = 3 mice for NT, n = 4 mice for GD2.CAR(I)IL12 and n = 5 mice for GD2.CAR.IL15 and GD2.CAR(I)IL12.TM NKTs, *p = 0.0112 GD2.CAR.IL15 vs GD2.CAR(I)IL12; *p = 0.0197 GD2.CAR.IL15 vs GD2.CAR(I)IL12.TM NKTs; log-rank Mantel−Cox test. Source data are provided as a Source Data file.

from each cycle of coculture and IFN-γ, IL-4 and IL-12 measured using a specific ELISA (R&D System). To quantify cytokines in mouse serum we used 15 μl of serum per sample and we proceeded according to the Luminex Performance assay Human Th1/Th2 Fixed Panel protocol (R&D system). The data were acquired on the Bio-Plex 200 System (BIO-RAD). Plasma obtained from mice was analyzed for the presence of human cytokines using the Luminex Performance Assay Human Th1/Th2 fixed panel (R&D Systems) following the manufacturer's instructions. Data were collected and analyzed using the Bio-Plex Manager 6.1 software (Bio-Rad).

## Xenotransplant mouse models

In the experiment to assess the persistence of NKTs in vivo, female and male NSG mice (7 - 9 weeks of age, obtained from the UNC Animal Core) were injected intravenously (i.v.) via tail injection with $1 \times 10^7$ GFP, IL12(I)GFP NKTs, CD62LΔNGFR or IL12(I)GFP KO.SELL. Mice were bled every 10 days to assess the persistence of NKTs. Mice were euthanized at day 30 and peripheral blood was collected from the heart, while spleen and liver were smashed on cell strainers and washed with 2 mL of PBS. Peripheral blood, spleen and liver were analyzed to detect the presence of NKTs by flow cytometry. At the time of sacrifice IL12(I)GFP NKTs were isolated from the spleen of 5 mice and purified from mouse cells using anti-human CD45 microbeads (Miltenyi Biotech). $3 \times 10^6$ IL12(I)GFP NKTs were injected i.v. in NSG mice and NKT persistence was monitored for 30 days. In the experiment to assess the anti-tumor activity of NKTs in vivo in the leukemia model, female and male NSG mice (7–9 weeks of age, obtained from the UNC Animal Core) were injected intravenously (i.v.) via tail injection with $1 \times 10^6$ Ffluc-labeled BV-173 tumor cells. Four days after tumor cell injection, mice were infused i.v. with $9 \times 10^6$ CAR.GD2(I)IL12, CAR.CD19 or CAR.CAR19(I)IL12 transduced NKTs. Tumor growth was monitored weekly by bioluminescence (BLI; total flux, photons/second) using the IVIS kinetic in vivo imaging system (PerkinElmer). Mice were sacrificed according to tumor growth (BLI above $5 \times 10^8$ photons/second), UNC guidelines for signs of discomfort (weight loss, lethargy or hunch), or to terminate the experiment. When mice were euthanized, peripheral blood was collected from the heart, while spleen and liver were smashed on cell strainers and washed with 2 mL of PBS. Peripheral blood, spleen and liver were analyzed to detect the presence of NKTs by flow cytometry. In the experiment to assess the anti-tumor activity of NKTs in vivo in the neuroblastoma model, female and male NSG mice (7–9 weeks of age, obtained from the UNC Animal Core) were injected intravenously (i.v.) via tail injection with $1 \times 10^6$ Ffluc-labeled CHLA-255 tumor cells. Seven days after tumor cell injection, mice were infused i.v. with $1 \times 10^7$ or $5 \times 10^6$ CAR.CD19(I)IL12, CAR.GD2 or CAR.GD2(I)IL12 transduced NKTs. Eight weeks after the NKTs injection, mice that were tumor free received a second injection i.v. of $1 \times 10^6$ FFluc-labeled CHLA-255 tumor cells. Neuroblastoma tumor growth was monitored weekly by bioluminescence (BLI; total flux, photons/second) using the IVIS kinetic in vivo imaging system (PerkinElmer). In selected experiments the weight of the mice was monitored weekly. Mice were sacrificed according to tumor growth (BLI above $6 \times 10^7$ photons/second), UNC guidelines for signs of discomfort (weight loss, lethargy or hunch), or to terminate the experiment. When mice were euthanized, peripheral blood was collected from the heart and spleen and liver were smashed on cell strainers and washed with 2 mL of PBS. Peripheral blood, spleen and liver were analyzed to detect the presence of NKTs by flow cytometry.

## RNA-seq and gene expression analysis

Briefly, total RNA was extracted from CAR-NKTs and messenger RNA libraries were prepared (TruSeq Stranded mRNA LibraryPrep, Illumina) and sequenced on the Illumina HiSeq4000 platform (UNC High-Throughput Sequencing Facility) using paired-end 100-bp reads, with 44 million reads on average (range, 1.5-149 million). RNA-seq data were aligned with STAR alignment (v.2.4.2) and quantified with Salmon (v.0.6.0). Paired-end FASTQ files were aligned to an Ensembl transcriptome (release 99, on reference genome GRCh38) using Star (v2.7.6a) and transcripts quantified using Salmon (v1.4.0). The quality of FASTQ data and quantified BAMs was verified using FastQC (v0.11.9) and Picard's (v2.22.4) CollectRnaSeqMetrics program, respectively. Differential gene expression was calculated and compared in R (v4.1.1) using the DESeq2 (v1.34.0) Bioconductor package.

## Statistical analysis

Data were summarized as the mean ± SD. Student's t test or two-way ANOVA were used to determine statistically significant differences between treatment groups, with Bonferroni's correction for multiple comparisons when appropriate (Prism 6: GraphPad Software). Survival analysis was performed using the Kaplan−Meier method (Prism 6: GraphPad Software). All P values less than 0.05 were considered statistically significant. For comparison of differential gene expression data in RNA sequencing, DESeq2[44] was utilized with default settings (Wald test) and adjusted for multiple hypothesis testing using DESeq2's implementation of the Benjamini-Hochberg method.

## Reporting summary

Further information on research design is available in the Nature Portfolio Reporting Summary linked to this article.

## Data availability

Source data for this study have been provided as source data file. RNA Seq data that support the findings of this study have been deposited in GEO with the accession code GSE241586 The remaining data are available within the Article, Supplementary Information or Source Data file. Source data are provided with this paper.

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

## Acknowledgements

The authors would like to thank Andrea Cosentino, Gene Therapy Program, Dana-Farber/Boston Children's Cancer and Blood Disorder Center for technical assistance in designing guided RNA. This work was supported by NIH National Cancer Institute R01-CA243543 (G.D) and R01-CA262250 (L.S.M). The PSC is supported in part by an NCI Center Core Support Grant (P30CA016086). The 10x Genomics Single-Cell RNAseq library preparation and sequencing were conducted by CGIBD's Advanced Analytics Core supported by NIH grant P30 DK034987. Figures 3a, 4e, 6a, 7f and S5 were created with Biorender.com.

## Author contributions

Conceptualization: E.L., B.S., L.M., and G.D.; Methodology: E.L., M.W., Gi.C., P.D., P.G. B.S., L.M., and G.D.; Investigation: E.L., M.W., G.B., Ga.C., V.C, S.S, H.H, L.F., T.W.; Formal Analysis: M.W.; Writing-Original Draft: E.L., M.W., and G.D.; Visualization: E.L., M.W., and G.D.; Supervision: B.S. and G.D.

## Competing interests

Dr. Dotti serves in the SAB of Catamaran Bio and Estella. Dr. Savoldo, Landoni and Dotti filed a patent for the technology developed in this manuscript (Title: Natural killer T-cells and methods of using the same. WO2023147564A3). No potential conflicts of interest were disclosed by the other authors in relation to this specific manuscript.
