## [Peer Review File · Nature Communications]

IL-12 Reprograms CAR-Expressing Natural Killer T cells to Long-Lived Th1-Polarized Cells with Potent Antitumor ActivityEditorial Note: Parts of this Peer Review File have been redacted as indicated to remove third-party material where no permission to publish could be obtained.

REVIEWER COMMENTS

Reviewer #1 (Remarks to the Author): with expertise in immune cell engineering, IL12

Landoni et al presents study that reports the role of IL-12 on promoting CAR NKT cell activity through IL-12 effect on inducing memory, Th1-polarization. The elevated CAR NKT activity was demonstrated against two different tumor models, targeting CD19 and GD2. In both animal models, the presence of IL12 had a profound effect on NKT expansion, retention of memory phenotype, persistence, and effector function. To avoid the potential systemic toxicity of IL-12 the authors also constructed membrane anchored IL-12, which resulted in CAR NKT cells exhibiting comparable activity as IL-12 secreting CAR while with no systemic IL-12. Experiments were designed to rigorously test the hypothesis, CAR NKT activity, and IL-12 influence.

One minor comment is about the possibility of membrane anchored IL12 to be released outside of cells. Prior study by Zhang et al (J Immunothera Cancer 2020:8) found the release of IL12 from cells when it was expressed as a fusion to CD8 transmembrane domain. IL12 release may be caused by proteolytic cleavage or be found within extracellular vesicles which may be produced when T cells are activated. Despite the use of same or similar transmembrane sequence in the current work, the authors did not detect systemic release of IL12 in animals. To test this possibility more rigorously, I suggest testing IL12 release in vitro after incubation of CAR NKT cells with target cells.

In Discussion, I suggest adding more in depth discussion on comparison of the effects of IL12 on NK or NKT cells with IL15, which is the cytokine tested in clinical trial with NK cells, and the potential advantage of CAR NKT over conventional T cells.

Reviewer #2 (Remarks to the Author): with expertise in NKT-cell based immunotherapy,

neuroblastoma

The manuscript by Landoni et al. shows that IL-12 reprograms NKT cells to Th1 polarized memory-like cells, thus adoptively transferred IL-12-CAR NKT cells retain for long periods and elicit strong antitumor immunity. Available data adequately support the critical statements of the study. The data are clear and interesting. However, it would be nicer if the following points were addressed to distinguish this study from previous work and give this paper a novelty.

- 1) The authors previously reported that anti-GD2 CAR NKT cells, which coexpress IL-15, show prolonged persistence and enhanced antitumor activity (Ref 6, 15). In this paper, GD2 CAR is used as a control of GD2.CAR(I)IL-12. Is GD2.CAR(I)IL-12 superior to GD2-CAR.15?
- 2) The authors also previously reported that IL-21 selectively protects CD62L+ NKT cells and enhances their effector function. Do IL-21 and IL-12 have a synergistic effect on reprogramming of NKT cells to Th1 memory-like cells?
- 3) Figure 1G; Phosphorylation of STAT4 is detected from the IL12(I)GFP lane. On the other hand, STAT4 expression in IL12(I)GFP is much lower compared to control samples. Does IL-12 signaling downregulate STAT4? Please address this point.
- 4) Page 11; Please check the concentration of α GalCer. In addition, the authors need to describe the source of α GalCer in the Methods section.

Reviewer #3 (Remarks to the Author): with expertise in NKT cells

The manuscript by Landoni and colleagues reports a protocol and data and the use of human iNKT cells engineered to produce IL-12p70 to fight solid tumors. The protocol appears well-controlled and robust leading to a long-term proliferation and survival in vivo associated with a memory phenotype, which is dependent on the IL-12/CD212 pathway. These iNKT cells undergo an important transcriptomic reprogramming associated with potent antitumoral activities and increased polyfunctionality as compared to control groups. In addition, co-expression of two validated CAR in IL-12-producing iNKT cells enhanced antitumoral activities both in vitro and in vivo. Finally, generation of iNKT cells that produce membrane-bound IL-12 displayed similar activities as compared to the soluble version that could prevent IL-12-driven toxicity. Thus, this paper provides important conceptual

advances in the field of iNKT cell-based immunotherapy. However, the following points could be taken into account to strengthen the message conveyed:

- An important point is the fact that the number of iNKT cells continues to increase in blood until day 30 in NSG mice. It would be of interest to monitor the number of iNKT cells at later time-points. Can NSG mice eliminate engineered iNKT cells? This is important for translation to humans.
- The Isoplexis experiment is of interest and indicate important heterogeneity in cytokine profiles in transduced human iNKT cells. However, it is difficult to appreciate the interindividual variability the way the data are presented. Can some cytokine profiles be further exploited to define iNKT cells with more potent antitumoral activities?
- The Isoplexis data also point towards some levels of heterogeneity and this could illustrate a caveat in the interpretation of the bulk RNA-Seq data. The use of a single-cell RNA-Seq approach might be more suitable to generate meaningful data in transduced human iNKT cells.
- It could be of interest to test other type of activation signals such as cytokines in order to compare this to anti-TCR mAb.
- How is the apoptosis rate in transduced iNKT cells as compared to controls?
- Authors used in some cases anti-Vb11 mAb to monitor iNKT cells instead of the mAb directed against the invariant alpha chain. Is there any reason for that? How is the Vb repertoire in transduced human iNKT cells?
- How is the proliferative rate of iNKT cells that de novo express CD62L as compared to iNKT cells that readily express iNKT cells?
- Can the authors discuss why not all iNKT cells acquire CD62L in vivo?
- It would be of interest to quantify soluble plasma factors that could be associated with liver and/or renal toxicity upon iNKT cell transfer? Is there any differences in the levels of these factors between iNKT cells producing membrane-bound IL-12 compared to soluble IL-12?

Minor points:

a-GalCer cannot be solely considered as a synthetic molecule.

Dear Reviewers,

We would like to thank the reviewers for their constructive comments on our manuscript. Based on the comments, we conducted additional experiments, generated new data, and revised the manuscript accordingly. In particular, we experimentally compared the activity of NKTs expressing either IL-12 or IL-15 *in vivo* taking into consideration that we have tested clinically CAR-NKTs co-expressing IL-15(1;2). These new data showed the superiority of the IL-12-based strategy. We also experimentally addressed the comparison *in vitro* of IL-12, IL-15, and IL-21 expressing NKTs showing that only IL-12 promotes upregulation of CD62L as a surrogate marker of the signature we have identified. Finally, we addressed the points related to potential IL-12 leakage, NKT persistence *in vivo* as well as safety/toxicities.

REVIEWER COMMENTS

Reviewer #1 (Remarks to the Author): with expertise in immune cell engineering, IL12

Landoni et al presents study that reports the role of IL-12 on promoting CAR NKT cell activity through IL-12 effect on inducing memory, Th1-polarization. The elevated CAR NKT activity was demonstrated against two different tumor models, targeting CD19 and GD2. In both animal models, the presence of IL 12 had a profound effect on NKT expansion, retention of memory phenotype, persistence, and effector function. To avoid the potential systemic toxicity of IL-12 the authors also constructed membrane anchored IL-12, which resulted in CAR NKT cells exhibiting comparable activity as IL-12 secreting CAR while with no systemic IL-12. Experiments were designed to rigorously test the hypothesis, CAR NKT activity, and IL-12 influence.

One minor comment is about the possibility of membrane anchored IL 12 to be released outside of cells. Prior study by Zhang et al (J Immunothera Cancer 2020:8) found the release of IL 12 from cells when it was expressed as a fusion to CD8 transmembrane domain. IL 12 release may be caused by proteolytic cleavage or be found within extracellular vesicles which may be produced when T cells are activated. Despite the use of same or similar transmembrane sequence in the current work, the authors did not detect systemic release of IL 12 in animals. To test this possibility more rigorously, I suggest testing IL 12 release in vitro after incubation of CAR NKT cells with target cells.

We tested the release of IL-12 *in vitro* by CAR-NKTs in response to tumor cells. As indicated by the reviewer and reported in **Supplementary Figure S10F**, when we compared CAR-NKTs expressing either the soluble form of IL-12 or the membrane-bound form of IL-12, IL-12 detection in the supernatant was significantly lower in CAR-NKTs expressing the membrane-bound IL-12. However, as suggested, we also tested the release of IL-12 *in vitro* in NKTs after iTCR stimulation and confirmed significantly lower IL-12 release from the membrane-bound version compared to the soluble IL-12 (**Revised Supplementary Figure S10C**). As suggested, IL-12 release in the case of the membrane-bound form of IL-12 may be caused by proteolytic cleavage or be found within extracellular vesicles released by activated NKTs. This effect is noticeable *in vitro*, while IL-12 is undetectable *in vivo* in the plasma of mice treated with NKTs expressing the membrane-bound form of IL-12, while IL-12 is clearly detectable in the plasma of mice treated with CAR-NKTs expressing the soluble form of IL-12. We cannot exclude that IL-12 is present in the plasma of mice treated with CAR-NKTs expressing the membrane-bound IL-12, but IL-12 is below the limit of detection of the assay.

In Discussion, I suggest adding more in depth discussion on comparison of the effects of IL 12 on

NK or NKT cells with IL15, which is the cytokine tested in clinical trial with NK cells, and the potential advantage of CAR NKT over conventional T cells.

We have now performed experiments *in vivo* in which we compared NKTs expressing either IL-12 or IL-15 (**Revised Figure 7** and **Supplementary Figure S10**). We added specific comments in the discussion concerning the comparison of the two cytokines.

Reviewer #2 (Remarks to the Author): with expertise in NKT-cell based immunotherapy, neuroblastoma

The manuscript by Landoni et al. shows that IL-12 reprograms NKT cells to Th1 polarized memory-like cells, thus adoptively transferred IL-12-CAR NKT cells retain for long periods and elicit strong antitumor immunity. Available data adequately support the critical statements of the study. The data are clear and interesting. However, it would be nicer if the following points were addressed to distinguish this study from previous work and give this paper a novelty.

1) *The authors previously reported that anti-GD2 CAR NKT cells, which coexpress IL-15, show prolonged persistence and enhanced antitumor activity (Ref 6, 15). In this paper, GD2 CAR is used as a control of GD2.CAR(I)IL-12. Is GD2.CAR(I)IL-12 superior to GD2-CAR.15?*

As indicated to **Reviewer 1**, we performed comparative experiments *in vivo* of CAR-NKTs expressing either soluble or membrane-bound IL-12 versus CAR-NKTs expressing IL-15 that we have used in the clinical study reported in Nature Medicine in patients with neuroblastoma (1;2). We have found superior antitumor activity of CAR-NKTs expressing IL-12 in the neuroblastoma model. These data have been included in the **Revised Figure 7** and **Supplementary Figure S10**.

2) *The authors also previously reported that IL-21 selectively protects CD62L+ NKT cells and enhances their effector function. Do IL-21 and IL-12 have a synergistic effect on reprogramming of NKT cells to Th1 memory-like cells?*

We performed comparative experiments *in vitro* of CAR-NKTs expressing either soluble or membrane-bound IL-12 versus CAR-NKTs expressing either IL-15 or IL-21. IL-21 did not polarize NKTs towards a Th1 phenotype nor we could observe a synergistic effect with IL-12. These data have been included in the **Revised Supplementary Figure S3**. Of note in the paper mentioned from the reviewer the cells were grown from day 0 in presence of recombinant IL-21, while our NKTs were transduced with a retrovirus to constitutively express IL-21.

3) *Figure 1G; Phosphorylation of STAT4 is detected from the IL12(I)GFP lane. On the other hand, STAT4 expression in IL12(I)GFP is much lower compared to control samples. Does IL-12 signaling downregulate STAT4? Please address this point.*

The reviewer is highlighting an interesting possibility. We re-analyzed the RNAseq data set, and we observed that STAT4 expression was significantly decreased in IL12(I)GFP samples (-0.29 log₂ fold-change, p<0.034 by Benjamini-Hochberg FDR correction). We also examined the expression of known STAT4 regulators such as SOCS3 and PIAS2 and observed a significant increase in SOCS3 (3.05 log₂ fold-change, p<8.90e-27), suggesting that IL-12 signaling may down-regulate STAT4 at the transcriptional level. The analyses is included here for the reviewer. However, we do not feel that this part should be included in the manuscript at the current stage, and we hope the reviewer agrees.

4) Page 11; Please check the concentration of α GalCer. In addition, the authors need to describe the source of α GalCer in the Methods section.

This comment has been addressed in the revised manuscript.

Reviewer #3 (Remarks to the Author): with expertise in NKT cells

The manuscript by Landoni and colleagues reports a protocol and data and the use of human iNKT cells engineered to produce IL-12p70 to fight solid tumors. The protocol appears well-controlled and robust leading to a long-term proliferation and survival in vivo associated with a memory phenotype, which is dependent on the IL-12/CD212 pathway. These iNKT cells undergo an important transcriptomic reprogramming associated with potent antitumoral activities and increased polyfunctionality as compared to control groups. In addition, co-expression of two validated CAR in IL-12-producing iNKT cells enhanced antitumoral activities both in vitro and in vivo. Finally, generation of iNKT cells that produce membrane-bound IL-12 displayed similar activities as compared to the soluble version that could prevent IL-12-driven toxicity. Thus, this paper provides important conceptual advances in the field of iNKT cell-based immunotherapy. However, the following points could be taken into account to strengthen the message conveyed:

1-An important point is the fact that the number of iNKT cells continues to increase in blood until day 30 in NSG mice. It would be of interest to monitor the number of iNKT cells at later time-points. Can NSG mice eliminate engineered iNKT cells? This is important for translation to humans.

We apologize if the presentation of the data was not clear. In **Figure S8B**, we quantified both GD2.CAR NKTs and GD2.CAR(I)IL12 NKTs in the peripheral blood at the time of sacrifice (day 40 and day 90, respectively). We hope the reviewer agrees that 90 days of observation in immunodeficient mice is quite long. Here we illustrate in the Figure below for the reviewer the direct comparison, and indeed GD2.CAR(I)IL12 NKTs expand further the longer they stay in the mice.

Since NSG mice do not have T cells or NK cells, human cells cannot be eliminated in this model. For the clinical translation, we do not anticipate the elimination of adoptively transferred CAR-NKTs expressing IL-12 in the autologous setting, unless an immune response can be triggered by the murine scFv of some CAR molecules. In contrast, we anticipate that CAR-NKTs expressing IL-12 will be rejected if used as allogenic products as observed for CAR-Ts without gene editing to remove MHC molecules.

2- The Isoplexis experiment is of interest and indicate important heterogeneity in cytokine profiles in transduced human iNKT cells. However, it is difficult to appreciate the interindividual variability the way the data are presented. Can some cytokine profiles be further exploited to define iNKT cells with more potent antitumoral activities?

We appreciate the reviewer's suggestion. We agree that the noted heterogeneity may be leveraged to further identify NKTs with more potent antitumor effects. IL12-expressing NKTs displayed significantly higher effector PSI. The effector group in this experiment is defined as cells that express Granzyme-B, IFN- γ , MIP-1 α , Perforin, TNF- α , and TNF- β . Unfortunately, these different subsets of polyfunctional NKTs cannot be segregated using known surface markers, and thus cannot be separated using conventional cell sorting. We are currently studying in depth how additional modifications of the CAR-NKTs expressing IL-12 could be used to enrich specific polyfunctional subsets. We hope the reviewer agrees that this could be an important follow-up project.

3-The Isoplexis data also point towards some levels of heterogeneity and this could illustrate a caveat in the interpretation of the bulk RNA-Seq data. The use of a single-cell RNA-Seq approach might be more suitable to generate meaningful data in transduced human iNKT cells.

This is an important suggestion and we are considering this possibility for the ongoing studies as discussed in the previous point. However, we hope the reviewer agrees that despite the limitations of the bulk RNAseq, the molecular signature of NKTs expressing IL-12 is striking taking into consideration that the signature identified in NKTs *in vitro* is maintained in NKTs persisting for 90 days in mice.

4- It could be of interest to test other type of activation signals such as cytokines in order to compare this to anti-TCR mAb.

We compared the activation of NKTs by either cross-linking the iTCR or cross-linking the CAR. As shown in the **Revised Figure S7C**, these two different stimulations provided similar effects. Moreover, CAR-NKTs were also activated by using tumor cell lines expressing the target antigen (**Figure 4D-G and Figures S6C, S7F and S9F**). These experiments also showed similar results.

5- How is the apoptosis rate in transduced iNKT cells as compared to controls?

We performed AnnexinV-7AAD staining of control and IL-12-modified NKTs and observed no differences between control NKTs and IL-12-modified NKTs after stimulation through the iTCR. These data have been included in the **Revised Figure S3B**.

6- Authors used in some cases anti-V β 11 mAb to monitor iNKT cells instead of the mAb directed against the invariant alpha chain. Is there any reason for that? How is the V β repertoire in transduced human iNKT cells?

We used the V β 11 antibody in selected experiments for convenience for the combination of antibodies with different fluorochromes. We feel confident using the anti-V α 24 and anti-V β 11 interchangeably since we previously reported that they are comparable to the iTCR pentamer in recognizing human NKTs(3). We have added this clarification in the Methods. Here we reproduce Figure 2B of the previous manuscript for comparison. In the same manuscript, we also sequenced the V β repertoire of NKTs, and confirmed that TRBV25-1 (V β 11) is the dominant beta chain used by human NKTs. Here we reproduce the Figure 3B of the previous manuscript to illustrate the V β repertoire of human NKTs.

Figure 2

[figure redacted]

Figure 3

[figure redacted]

6- How is the proliferative rate of iNKT cells that *de novo* express CD62L as compared to iNKT cells that readily express iNKT cells?

We compared the proliferative rate of NKTs expressing *de novo* CD62L versus NKTs expressing CD62L when isolated from the peripheral blood. We observed similar proliferative rates. These data have been included in the **Revised Figure S2F**.

7- Can the authors discuss why not all iNKT cells acquire CD62L *in vivo*?

As noted by the reviewer in a previous comment, human NKTs are heterogeneous. Not all CD62L negative cells exposed to IL-12 *in vitro* express CD62L, and similarly not all NKTs persisting *in vivo* show high CD62L expression. This could be further explored in the future. Furthermore, as we have demonstrated with SELL KO experiments, CD62L is not essential to maintain long-lived Th1 NKTs, it is not surprising that some long-lived Th1 NKT do not express CD62L.

8- It would be of interest to quantify soluble plasma factors that could be associated with liver and/or renal toxicity upon iNKT cell transfer? Is there any differences in the levels of these factors between iNKT cells producing membrane-bound IL-12 compared to soluble IL-12?

We have quantified plasma factors associated with liver (ALT and AST) and renal (Urea nitrogen and creatinine) toxicities at the time of sacrifice in mice treated with GD2.CAR(i)IL12 or GD2.CAR(i)IL12.TM according to the schema in **Figure 7F**. As a control, we used untreated mice (Ctr). As illustrated in the Figure below, we did not detect differences in treated mice compared to control mice.

However, we do not feel that these data should be included in the manuscript. According to Zou J. et al.(4), human IL-12 may not fully cross-react with murine cells, so these results may not fully represent the effects of IL-12. We hope the reviewer agrees.

Minor points:

We removed “synthetic” form the text.

Reference List

1. Heczey,A., Courtney,A.N., Montalbano,A., Robinson,S., Liu,K., Li,M., Ghatwai,N., Dakhova,O., Liu,B., Raveh-Sadka,T. et al 2020. Anti-GD2 CAR-NKT cells in patients with relapsed or refractory neuroblastoma: an interim analysis. *Nat. Med.* **26**:1686-1690.
2. Heczey,A., Xu,X., Courtney,A.N., Tian,G., Barragan,G.A., Guo,L., Amador,C.M., Ghatwai,N., Rathi,P., Wood,M.S. et al 2023. Anti-GD2 CAR-NKT cells in relapsed or refractory neuroblastoma: updated phase 1 trial interim results. *Nat. Med.*
3. Landoni,E., Smith,C.C., Fuca,G., Chen,Y., Sun,C., Vincent,B.G., Metelitsa,L.S., Dotti,G., and Savoldo,B. 2020. A High-Avidity T-cell Receptor Redirects Natural Killer T-cell Specificity and Outcompetes the Endogenous Invariant T-cell Receptor. *Cancer Immunol. Res.* **8**:57-69.
4. Zou,J.J., Schoenhaut,D.S., Carvajal,D.M., Warriar,R.R., Presky,D.H., Gately,M.K., and Gubler,U. 1995. Structure-function analysis of the p35 subunit of mouse interleukin 12. *J Biol. Chem.* **270**:5864-5871.

REVIEWERS' COMMENTS

Reviewer #1 (Remarks to the Author):

The authors fully addressed in their revised version concerns and questions raised from the prior review.

Reviewer #2 (Remarks to the Author):

Thank you to the authors for their careful revision of their manuscript. The authors have responded appropriately to the comments made by the reviewers, and the manuscript has been improved.

Reviewer #3 (Remarks to the Author):

Dear,

I would like to thank the authors for their efforts in discussing reviewer's comments including mine as well as in providing new data.

This significantly strengthen the message of the manuscript that is is to my opinion suitable for publication in Nat Comms.

Regards